# De novo histidine biosynthesis protects *Mycobacterium tuberculosis* from host IFN-γ mediated histidine starvation

Abhisek Dwivedy [1,3], Anam Ashraf [1,3], Bhavya Jha [1], Deepak Kumar [1], Nisheeth Agarwal[2] & Bichitra K. Biswal [1✉]

Intracellular pathogens including *Mycobacterium tuberculosis* (*Mtb*) have evolved with strategies to uptake amino acids from host cells to fulfil their metabolic requirements. However, *Mtb* also possesses de novo biosynthesis pathways for all the amino acids. This raises a pertinent question- how does *Mtb* meet its histidine requirements within an in vivo infection setting? Here, we present a mechanism in which the host, by up-regulating its histidine catabolizing enzymes through interferon gamma (IFN-γ) mediated signalling, exerts an immune response directed at starving the bacillus of intracellular free histidine. However, the wild-type *Mtb* evades this host immune response by biosynthesizing histidine de novo, whereas a histidine auxotroph fails to multiply. Notably, in an IFN-γ$^{-/-}$ mouse model, the auxotroph exhibits a similar extent of virulence as that of the wild-type. The results augment the current understanding of host-*Mtb* interactions and highlight the essentiality of *Mtb* histidine biosynthesis for its pathogenesis.

[1] National Institute of Immunology, New Delhi, India. [2] Translational Health Science and Technology Institute, NCR Biotech Science Cluster, Faridabad, India. [3] These authors contributed equally: Abhisek Dwivedy, Anam Ashraf. ✉email: bbiswal@nii.ac.in

Tuberculosis (TB), caused by *Mycobacterium tuberculosis* (*Mtb*), remains a major health issue worldwide[1]. This is primarily attributed to the ability of the pathogen to subvert host anti-bacterial immunity, its persistent nature, failure of the current drug regimen to completely eliminate the bacillary load and the emergence of drug-resistant strains[2,3]. In order to develop novel approaches to curtail *Mtb* infection, further elucidation of the host defence mechanisms and the countermeasures employed by *Mtb* is pertinent. A fundamental strategy that host employs to prevent the proliferation of an intracellular pathogen is by starving the bug of essential resources such as amino acids. For example, host limits *Chlamydia* and *Leishmania* infections by starving them of the amino acid tryptophan. Since these organisms are natural tryptophan auxotrophs, this strategy is effective in clearing the infections[4,5]. However, *Mtb* evades such immune strategy of the host by biosynthesizing tryptophan de novo[6]. The advent of the much awaited complete genome sequence of *Mtb* (strain H37Rv) in 1998 revealed that the pathogen possesses enzymes for the biosynthesis of all the 20 amino acids[7]. Owing to the proteinogenic functions of amino acids, the enzymes regulating their biosynthesis are viable anti-TB drug targets, particularly those involved in the synthesis of essential amino acids[6,8–11]. Contrary to a common perception that *Mtb* may acquire amino acids including methionine and tryptophan from the host, studies have demonstrated that the disruption of their biosynthesis pathways independently inhibits the bacterial proliferation[6,8,12], suggesting that the de novo biosynthesis of these amino acids is important. Histidine, an essential amino acid, regulates a wide range of cellular processes in addition to its proteinogenic and catalytic functions. Histidine is utilised in the biosynthesis of key metabolites such as histamine, carnosine[13] and thyrotropin releasing hormone[14]. In addition, histidine plays key roles in the maintenance of the cellular pH and metal chelation[15]. While histidine is synthesised de novo in plants and bacteria, the histidine catabolic pathways are found in all kingdoms of life (Supplementary Fig. 1a). Given the premise that the function of histidine is multifaceted and the enzymes involved in its biosynthesis are regarded as potential anti-TB drug targets[9,16,17], we sought to investigate – how does *Mtb* manage its histidine requirement in both in vitro and in vivo settings? We observed that in vitro growth of *Mtb* is dependent on de novo synthesis of histidine, as histidine supplementation is essentially required when the biosynthesis pathway is disrupted. However, the scenario is interestingly different in an in vivo mouse model of infection. We find that 2 weeks post infection, the host through an interferon gamma (IFN-γ) mediated mechanism upregulates its histidine catabolizing enzymes – histidine ammonia-lyase (HAL) and histidine decarboxylase (HDC), possibly to starve the *Mtb* of free intracellular histidine. However, the wild-type *Mtb* is able to kick-start the de novo biosynthesis of histidine to sustain growth, but a histidine auxotroph fails to grow. Notably, the same auxotroph grows well, similar to the wild-type *Mtb* in an IFN-γ$^{-/-}$ mouse model. Collectively, our study furthers current insights into the host–pathogen interactions associated with the mechanism of histidine fulfilment by *Mtb*.

## Results

### Histidine auxotrophy in *Mtb* is bactericidal and hinders pathogenesis.
To assess the extent of essentiality of histidine biosynthesis in *Mtb* growth, we compared the growth kinetics of *Mtb* H37Rv (H37Rv), Δ*hisD* (*Mtb* H37Rv strain with a deletion mutation in the *hisD* gene) and *chisD* (*hisD* gene externally complemented into Δ*hisD*; Supplementary Fig. 1b) strains (Fig. 1a, b). While the H37Rv strain exhibited a normal growth

(red line), the Δ*hisD* exhibited a growth arrest over a period of 2 weeks (orange line). The addition of external histidine on day 7 (blue line) failed to restore the growth of Δ*hisD* as compared to the addition of external histidine on day 1 (lime line). This suggests the bactericidal nature of the auxotrophy. These results are in concordance with the previously reported data comparing the growth profiles of H37Rv and Δ*hisD*[18,19]. The *chisD* however showed a growth profile comparable to that of H37Rv. Bactericidal mutations in pathogenic bacteria are known to exert adverse effects on the natural course of their pathogenesis. However molecular histidine is abundantly present in mammalian cells (mouse plasma contains ~33 μM of histidine, totalling up to ~10 μg in 2 ml of mouse blood)[20,21], suggesting that *Mtb* may never have to experience histidine auxotrophy in its natural environment within the host. In addition, recent studies have demonstrated the various mechanisms by which *Mtb* captures and utilises host amino acids while residing within the phagosomes[22,23]. Nonetheless, we determined the survivability of H37Rv, Δ*hisD* and *chisD* strains within mouse and human monocytic cell lines, Raw264.7 and THP1, respectively. As expected, our results revealed a significant reduction in bacterial counts for Δ*hisD* compared to H37Rv and *chisD* over a period of 48 h post infection in both the cell lines (Supplementary Fig. 1c). A similar reduction in bacillary counts of Δ*hisD* was also observed in primary macrophages isolated from H37Rv-infected mice (Fig. 1c). This indicates that histidine auxotrophy adversely affects the pathogenesis of *Mtb* in an ex vivo setting.

To delineate the effects of histidine auxotrophy under in vivo infection conditions, C57BL/6 (B6) mice were infected with H37Rv, Δ*hisD* and *chisD* and the bacillary loads in lung tissues were determined on days 1, 15, 21, 28 and 63 post infection. As shown in Fig. 1d, irrespective of the minimal difference in the inoculum as observed on day 1, the bacillary loads were found nearly similar for all the three strains on day 15. However, a reduction in bacterial counts of Δ*hisD* was observed on day 21 and the trend continued till day 63, when the counts for Δ*hisD* were found nearly a 100-fold lesser compared to either H37Rv or *chisD*. The bacterial counts for the H37Rv or *chisD* remained nearly stable on days 28 and 63 (Fig. 1d). A similar decline in the bacillary loads in the spleens of infected mice was observed with Δ*hisD*-infected spleens displaying almost negligible bacterial load on day 63 (Supplementary Fig. 1d). Although, significant lesions were observed in lungs on day 28 of infection with H37Rv, Δ*hisD* and *chisD*, these were more pronounced in mice infected with H37Rv and *chisD* (Fig. 1e). On day 63, the Δ*hisD*-infected mice lungs were nearly as healthy as that of uninfected mice (Fig. 1e). On days 28 and 63, lungs from all the mice infected with H37Rv and *chisD* exhibited higher number of lesions and reduced air spaces compared to those from Δ*hisD*-infected animals (Fig. 1f). Together, these results indicate a possible contribution of *Mtb* HisD and the de novo histidine biosynthesis towards a successful in vivo *Mtb* infection.

### A concomitant up-regulation of *Mtb* histidine biosynthesis enzymes and host histidine sequestering enzymes.
In order to delineate the role of bacterial histidine biosynthesis in *Mtb* infection, we examined the dynamics of cellular expression of the histidine biosynthesis pathway enzymes as the infection progresses. For this, three key enzymes of the pathway – imidazole glycerol phosphate dehydratase (*hisB*, Rv1601)[24], histidinol dehydrogenase (*hisD*, Rv1599)[18] and histidinol phosphate phosphatase (*hisN*, Rv3137)[25,26], well known for their essentiality were chosen. These enzymes were overexpressed and purified to homogeneity using appropriate protocols (Supplementary Fig. 2a, b) and specific polyclonal antibodies were raised in rabbit. The

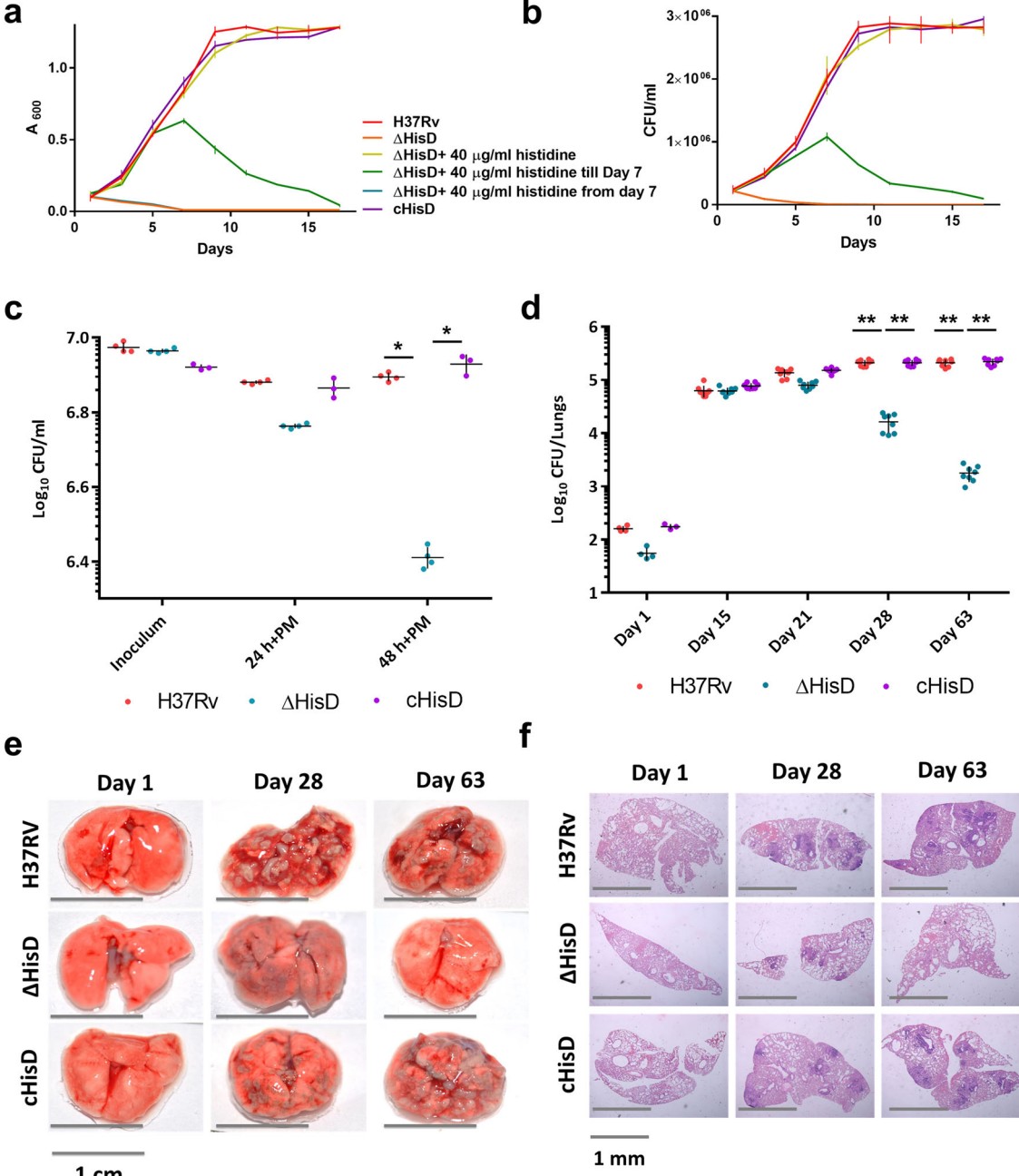

**Fig. 1 Histidine auxotrophy is bactericidal and impedes pathogenesis of *Mtb*. a** Comparison of the growth kinetics of H37Rv, Δ*hisD* and c*hisD* in the presence and absence of external histidine shows the inability of the Δ*hisD* to multiply in the absence of external histidine supplementation ab initio ($n = 9$ independently grown bacterial cultures; mean and SEM). **b** Bacillary counts corresponding to the growth kinetics show a decline of Δ*hisD* when no external histidine is supplemented ab initio ($n = 9$ independently grown bacterial cultures; mean and SEM). **c** A decline in bacterial counts of Δ*hisD* (>5-fold) over a course of 48 h post infection in Primary macrophages (PM); completely rescued in c*hisD* hint at the indispensability of HisD in ex vivo infection ($n = 4$ independently acquired and cultured primary cells; mean and SEM; *$P$-value < 0.05). **d** Bacillary load in B6 mice lungs over a period of 9 weeks depicting ~10 and 100-fold declines in bacterial counts of Δ*hisD* at days 28 and 63, respectively, compared to H37Rv and c*hisD* ($n = 8$ individual mice per time point; mean and SEM; **$P$-value < 0.005). **e** The disease rapidly progresses in H37Rv and c*hisD* infections, as is evident from a large number of lesions on the external physiology of the lungs. Δ*hisD* shows minimal infection at day 28 and almost clears out by day 63 (pictures are representative of nine individual mice lungs samples). **f** Histopathology of lungs depicting decreased air spaces and higher number of lesions on days 28 and 63 in H37Rv and c*hisD*-infected mice. Δ*hisD* infected lungs have fewer granulomatous formations by day 28 with almost clean lungs on day 63 (pictures are representative of nine individual mice lungs samples).

antibodies exhibited high level of specificity for their respective antigens in the total mycobacterial cell lysate (Supplementary Fig. 2c). The affinities of these antibodies were determined using different concentrations of the purified enzymes (Supplementary Fig. 2d). Subsequently, the expression levels of these enzymes in

the whole lung lysates of H37Rv-infected B6 mice were examined through immunoblotting at successive intervals for the period of 2 months of infection. The change in the expression levels of histidine biosynthesis enzymes could possibly be dependent on the varying bacillary load. Thus, the levels of HisB, HisD and

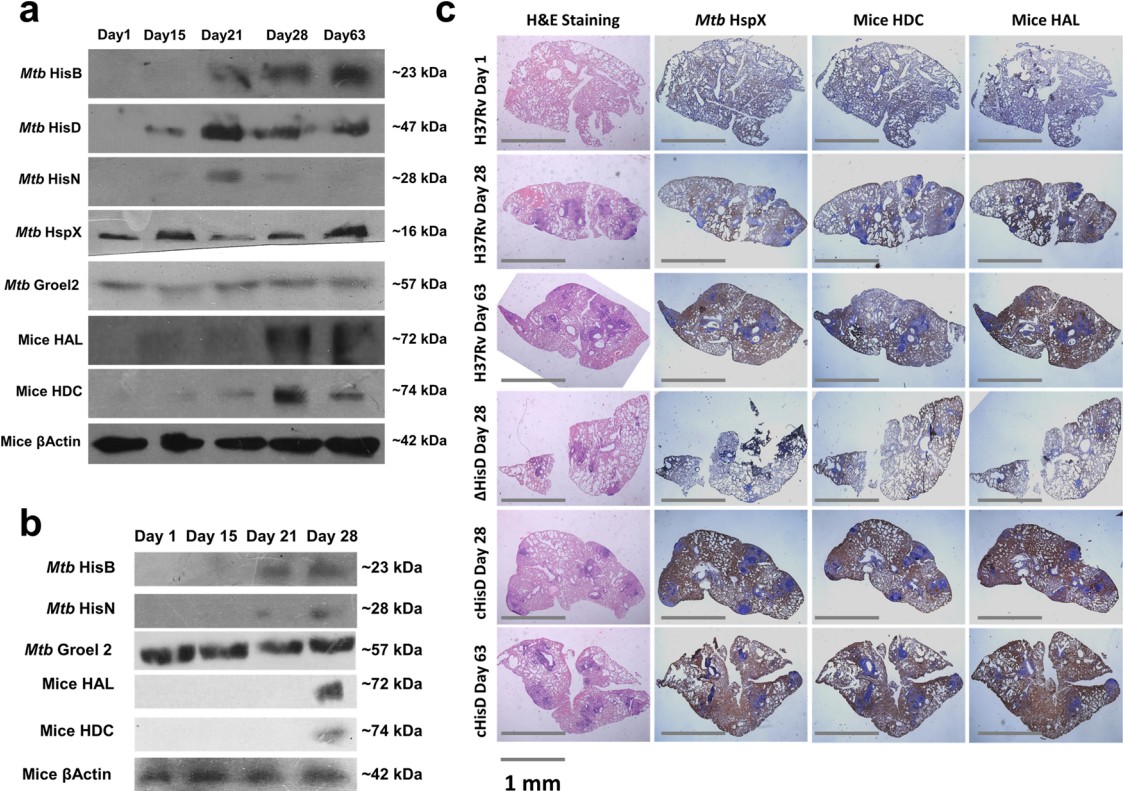

**Fig. 2 A concomitant increase in host histidine sequestering enzymes and *Mtb* histidine biosynthesis enzymes. a** Immunoblots depict a noticeable rise in the expression levels of histidine catabolizing enzymes HAL and HDC in the whole lungs lysate sample of H37Rv-infected B6 mice over a period of 9 weeks. A similar rise in the *Mtb* histidine biosynthesis enzymes HisB, HisD and HisN is also observed. Mice β-Actin and *Mtb* Groel2 serve as loading controls. *Mtb* HspX serve as an indicator of infection. (Immunoblots are representative of samples obtained from nine mice for each time point). **b** Immunoblots depicting a comparable rise in the expression levels of host histidine catabolizing enzymes and *Mtb* histidine biosynthesis enzymes in whole lungs lysate samples of Δ*hisD*-infected B6 mice over a period of 4 weeks (immunoblots are representative of samples obtained from nine mice each time point). **c** Immunohistochemical analysis exhibiting co-localised expression of mice HAL, mice HDC and *Mtb* HspX suggesting compartmentalised expression of host histidine sequestering enzymes limited to lung lesions. (Expression is depicted in bright blue colour. Pictures are representative of samples obtained from nine mice for each time point).

HisN proteins were determined after normalisation with the expression of *Mtb* Groel2, whereas the *Mtb* HspX was used as a marker of infection. Representative Ponceau-stained nitrocellulose membrane as indicators of equal loading are presented in Supplementary Fig. 3a. We observed increased expression levels of these enzymes on days 21, 28 and 63 post infection (Fig. 2a; HisB, HisD and HisN). Analysis of the aforementioned lung lysates by ELISA verified the expression dynamics of these enzymes as observed in immunoblots. We observed ~7-, 5- and 9-fold increase in the expression levels of HisB, HisD and HisN, respectively, on day 21 post infection (Supplementary Fig. 3b; HisB, HisD and HisN). We also noted a rise in the expression levels of HisB and HisN in whole lung homogenates of Δ*hisD*-infected mice on days 21 and 28 (Fig. 2b and Supplementary Fig. 3d; HisB & HisN). Representative Ponceau-stained nitrocellulose membrane as indicators of equal loading are presented in Supplementary Fig. 3c. This suggests that the Δ*hisD* fails to maintain a successful infection primarily due to the disruption in the histidine biosynthesis caused by the absence of a single enzyme- HisD. Since, histidine is abundant in the host cellular milieu[20,21] which is readily available for the bacterial uptake, we sought to determine why does *Mtb* activate its de novo histidine biosynthesis, which is an energy-driven process. We reasoned that the host might exert a defence mechanism to restrict histidine availability and making *Mtb* unable to import host-derived histidine.

In order to test our hypothesis, we examined the KEGG Pathways[27] for the probable histidine sequestering/catabolizing enzymes in mammals. The results suggested two enzymes involved in the catabolism of histidine- HAL and HDC. HAL catalyses the conversion of histidine to ammonia and urocanate and HDC catalyses the decarboxylation of histidine to histamine[28]. While both the aforementioned enzymatic reactions are reversible in nature, the *Mtb* genome[7] does not encode for HAL and HDC homologues. This implies that the bacillus is likely not capable of regenerating histidine from either urocanate or histamine. An increase in the expression levels of HAL and HDC was observed 3 weeks post infection in H37Rv-infected B6 whole lungs lysates. This corroborates with the dynamics of the expression of the histidine biosynthesis enzymes (Fig. 2a; HAL and HDC). ELISA of the aforementioned samples also showed nearly 2- and 3-folds up-regulation of both the enzymes on days 21 and 28, respectively (Supplementary Fig. 3b; HAL and HDC). A similar level of increase in the expression of HAL and HDC were observed in lung homogenates of Δ*hisD*-infected mice on day 28 (Fig. 2b and Supplementary Fig. 3d; HAL and HDC). The increase in the expression of HDC is often correlated to an increased production of histamine, a well-known mediator of inflammation[29,30]. However, the concurrent increase in the expression of HAL suggests an eventual decrease in the levels of intracellular free histidine (simplified as "free histidine" henceforth) within the host cellular milieu. It still remained unclear if this increased expression of HAL

and HDC occurred globally throughout the lung tissue or specifically in the bacilli filled lung lesions. Immunohistochemical analysis was carried out to decipher the localisation of the HAL and HDC expression. Interestingly, the expression profiles of HAL and HDC co-localised with the expression profiles of HspX, indicating that the expression of these enzymes were restricted specifically to the lung lesions (Fig. 2c). This further suggests that the up-regulation of HAL and HDC is primarily associated with sequestration of free histidine. It is well known that the adaptive immune responses activate in ~15–20 days post infection in a mouse model of TB[31,32]. Thus, together these results also hint towards a possible adaptive immune mechanism regulating the histidine sequestration event possibly aimed at an effective clearance of the bacilli.

**A host immune mediated signalling regulates the expression of HAL and HDC.** To determine any possible correlation between the levels of host histidine catabolism enzymes and the host immune signalling, the total mRNA extracted from whole lungs of uninfected and H37Rv-infected mice at 28 days post-infection was subjected to comparative transcriptomics analysis using next-generation RNA sequencing (RNASeq). The RNASeq data were analysed for their quality control (Supplementary Fig. 4a and Supplementary Tables 1 and 2), alignments (Supplementary Fig. 4b and Supplementary Table 3), feature counts (Supplementary Fig. 4c and Supplementary Table 4) and expression distribution (Supplementary Fig. 4d) by using using fastp[33] and MultiQC[34], Hisat2[35], FeatureCounts[36] and DeSeq2[37], respectively. The normalised read counts obtained from RNASeq are presented in Supplementary Data 1. A comparison between the total readouts from the infected and uninfected samples revealed 4066 and 3506 unique readouts, respectively. A total of 1477 unique transcripts were found to exhibit differential expression in the infected samples of which 545 were upregulated and 932 were downregulated. Contrary to these, the expression of 18974 transcripts remained neutral (Supplementary Data 2). These results thus clearly indicate that the RNASeq data are of optimum quality and can be employed for further analysis.

The differentially expressed genes were further analysed on Cytoscape 3.7.2[38,39]. Overrepresentation analysis of the differentially expressed genes using KEGG[40,41] and Reactome[42] databases resulted in an enrichment of pathways that belong to "Cytokine-cytokine receptor interactions", "Cytokine signalling in immune system", "Signalling by interleukins" with highest significance (Supplementary Figs. 5 and 6). We also examined the possible immune functions related to the differentially expressed gene set. A gene ontology (GO)- Immune Functions analysis using ClueGO[43] showed a significant enrichment of 4 GO terms-GO:0045089 (positive regulation of innate immune response), GO:0050863 (regulation of T cell activation), GO:0034341 (response to IFN-γ) and GO:0002478 (antigen processing and presentation of exogenous peptide antigen; Supplementary Fig. 7a, b and Supplementary Data 3). Of these, 22% of the transcripts were found related to IFN-γ responses. Further, a GO interaction network displayed significant interactions of the IFN-γ responses with the innate immunity and the T cell activation (Supplementary Fig. 8). The major cytokines within the aforementioned interaction network were IFN-γ, IL-6, IL-1b, IL-21, IL-2Ra and IL-27. In addition to those specific to the immune system, the interaction network also exhibited the presence of transcripts encoding proteins that are involved in cellular metabolism, apoptosis, transcriptional control and post-translational modifications (Ido1, Foxp3, Icos, Irf1, Irf7, Stat1, Pycard, Ubd, Rac1 and Dicer1 to name a few). Taken together, the interaction network reveals an extensive crosstalk between the cytokines and various non-immune signalling molecules, besides interaction among proteins involved in the overall immune related functions.

We further explored the key cytokine signalling pathways reportedly active during an acute TB infection. The differentially expressed transcripts significantly populated the IFN-γ signalling pathway from the Reactome Database[42] and WikiPathways[44], respectively (Supplementary Figs. 9 and 10; overlaid with their respective expression values). Amongst the other anti-mycobacterial cytokine pathways, the IL-6 pathway from the Reactome Database was also significantly enriched (Supplementary Fig. 11; respective expression values overlaid). Previous studies have shown that IFN-γ, TNF-α and IL-6 are the major anti-mycobacterial cytokines and IFN-γ is the central regulator of all cytokine signalling in an acute *Mtb* infection[45–47]. The primary and secondary responses for IFN-γ and IL-6 activate the JAK-STAT signalling pathways eventually leading to the expression of interferon regulatory factors (IRFs)[48,49]. The IRFs act as transcription factors and regulate the expression of a plethora of immune and metabolic genes, thus providing feedback control over type I and type II interferons[45,48,50–53]. Interestingly, previous studies suggest the presence of transcription factor binding sites for IRFs, STATs and NFκB upstream of the genes encoding for HAL and HDC[54]. The list of genes populating the IFN-γ and IL-6 signalling pathways (Supplementary Figs. 9–11) along with HAL and HDC (Supplementary Table 5) were then used to create a functional interaction network (Fig. 3a) using GeneMANIA[55,56] and Diffany[57]. This network depicts regulatory functions, co-localisations, co-expression, protein–protein interactions and predicted interactions. The network proposes both regulatory and predicted interactions of the IFN-γ and IL-6 signalling with the HAL and HDC. However, it is still unclear as to how does NFκB regulate HAL?

To validate the cellular expression of the signalling molecules from the aforementioned interaction network, we determined the expression levels of a set of randomly selected molecules using immunoblotting and ELISA of the H37Rv-infected lung lysates (Fig. 3b and Supplementary Fig. 12). The dynamics of STAT1 and STAT3 suggests an interesting scenario. STAT1 and STAT3 bind to the same receptor complex and have been shown to exhibit competitive binding[58]. In an ongoing infection, STAT3 is activated during the innate immune response, primarily through type I IFNs. STAT1, however, is activated by adaptive immune mediated IFN-γ through a JAK2 signalling cascade[59,60]. While STAT1 primarily controls the pro-inflammatory response resulting in pathogen clearance and apoptosis, STAT3 is involved in both pro- and anti-inflammatory signalling cascades. It is worth mentioning that STAT1 knockout mice are highly susceptible to mycobacterial infections[61]. In addition, production of IL-6 also suggest the onset of an adaptive immune response, often linked to the progression of a diseases from acute to chronic stages[62]. Remarkably, the expression profiles of the *Mtb* histidine biosynthesis and the host histidine catabolism enzymes correlate well with the expression of various signalling molecules and transcription factors (Fig. 3b). The transcriptomic analysis suggests that the IFN-γ signalling initiates the up-regulation of the HAL and HDC, possibly as a secondary immune response through the IRF/STAT transcriptional control (Supplementary Fig. 13). Taken together, these findings also indicate that the triggering of the histidine biosynthesis represents an *Mtb* counter response to host adaptive immunity directed at starving the bacilli of free histidine.

**An interferon gamma mediated clearance of the *Mtb* histidine auxotroph.** Previous studies suggest that the protective immunity in acute TB is provided by activated CD4 T cells[63]. Following the

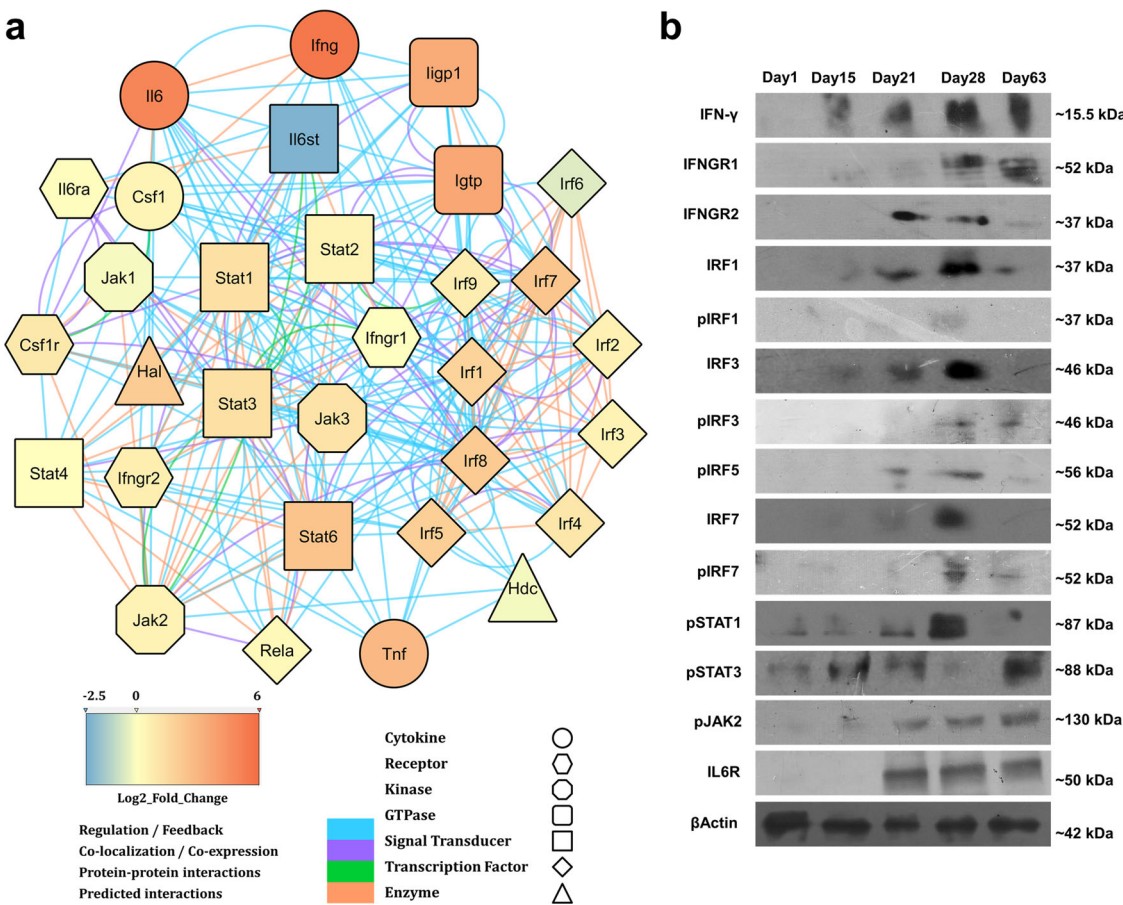

**Fig. 3 An adaptive immune driven signalling network regulates the expression of host histidine sequestering enzymes. a** An interaction network based on regulatory control, co-expression, co-localisation, protein–protein interactions and predicted interactions suggesting a possible regulation of HAL and HDC by IFN-γ signalling via the JAK-STAT pathway and the Interferon Regulatory Factors. (Fold changes obtained from RNAseq analysis are overlaid onto the respective molecule using the provided colour code. The nature of molecule and the nature of interaction are represented in shape and colour respectively.) **b** Immunoblots of mice lungs lysates reveal the expression of IFN-γ and selected downstream genes such as IRFs, JAK and STAT with disease progression. Mice β-Actin serves as the loading control (immunoblots are representative of samples obtained from nine mice for each time point).

infection, the bacilli reside in the vacuolar and cytoplasmic compartments of the alveolar macrophages and over the course of pathogenesis, a complex lesion is developed primarily to prevent bacillary escape[64–68]. An influx of a wide range of adaptive immune cells is observed within 2 weeks of infection[31,32]. Studies have also established that the CD4 T cells are particularly effective in clearing the bacillary load, employing a range of cytokines such as IFN-γ, TNF-α, IL-6 and IL-2[69–71]. We aimed to delineate a correlation, if any, between the declined bacillary counts of ΔhisD to that of an adaptive immune component. Primary macrophages infected with H37Rv and ΔhisD were stimulated with CD4 or CD8 T cells, both isolated from H37Rv-infected mice or naïve CD4 T cells isolated from uninfected mice. A significantly higher reduction in bacillary counts of ΔhisD was observed 48 h post infection following stimulation with CD4 T cells as compared to un-stimulated macrophages (Fig. 4a). Stimulation with CD8 T or naïve CD4 T cells however, did no show any effect on the bacillary counts of ΔhisD 48 h post infection (Supplementary Fig. 14a, b). These findings intrigued us to further examine the infectivity of ΔhisD in mice devoid of a CD4 T cell response. To explore this, C57BL/6-MHC-II knockout mice (B6 MHC-II−/−) were infected with H37Rv, chisD and ΔhisD, along with a control infection in B6 mice. The infected lungs of B6 mice displayed a 10-fold decline in bacillary counts for ΔhisD compared to H37Rv on day 28 post infection previously in this study (Fig. 1d). The

lungs of the B6 MHC-II−/− however displayed comparable bacillary loads for all the three strains (Fig. 4b). Similarly, B6 MHC-II−/− mice infected with H37Rv, ΔhisD and chisD, displayed near identical levels of lung damage as against B6 mice exhibiting more pronounced lung tissue damage upon infection with H37Rv and chisD compared to ΔhisD on day 28 post-infection (Fig. 4c). Overall, these results indicate the possible involvement of a CD4 T cell mediated mechanism in the reduction of ΔhisD bacterial loads.

To substantiate the role of any CD4 T cell effector cytokines involved in the bacillary clearance, we explored the expression levels of IFN-γ, which is reported as the most effective anti-tubercular cytokine[63,69]. As observed in the RNASeq data and its validation, there is a continuous presence of IFN-γ within the lung from day 15 to day 63 of infection (Supplementary Table 5 and Fig. 3b, IFN-γ). These results correspond with the bacillary counts, which remain similar for H37Rv, ΔhisD and chisD till 15 days post infection and the decline in ΔhisD was visible only from day 21 onwards (Fig. 1d). Also, an increase in the secretory load of IFN-γ was detected in the culture medium 48 h post infection (~1500 pg ml−1) in Mtb infected macrophages stimulated with CD4 T cell (Fig. 4d). To validate the efficacy of IFN-γ as the primary effector cytokine in the clearance of ΔhisD, primary macrophages and monocytic cell lines infected with H37Rv and ΔhisD were treated with an external dose of

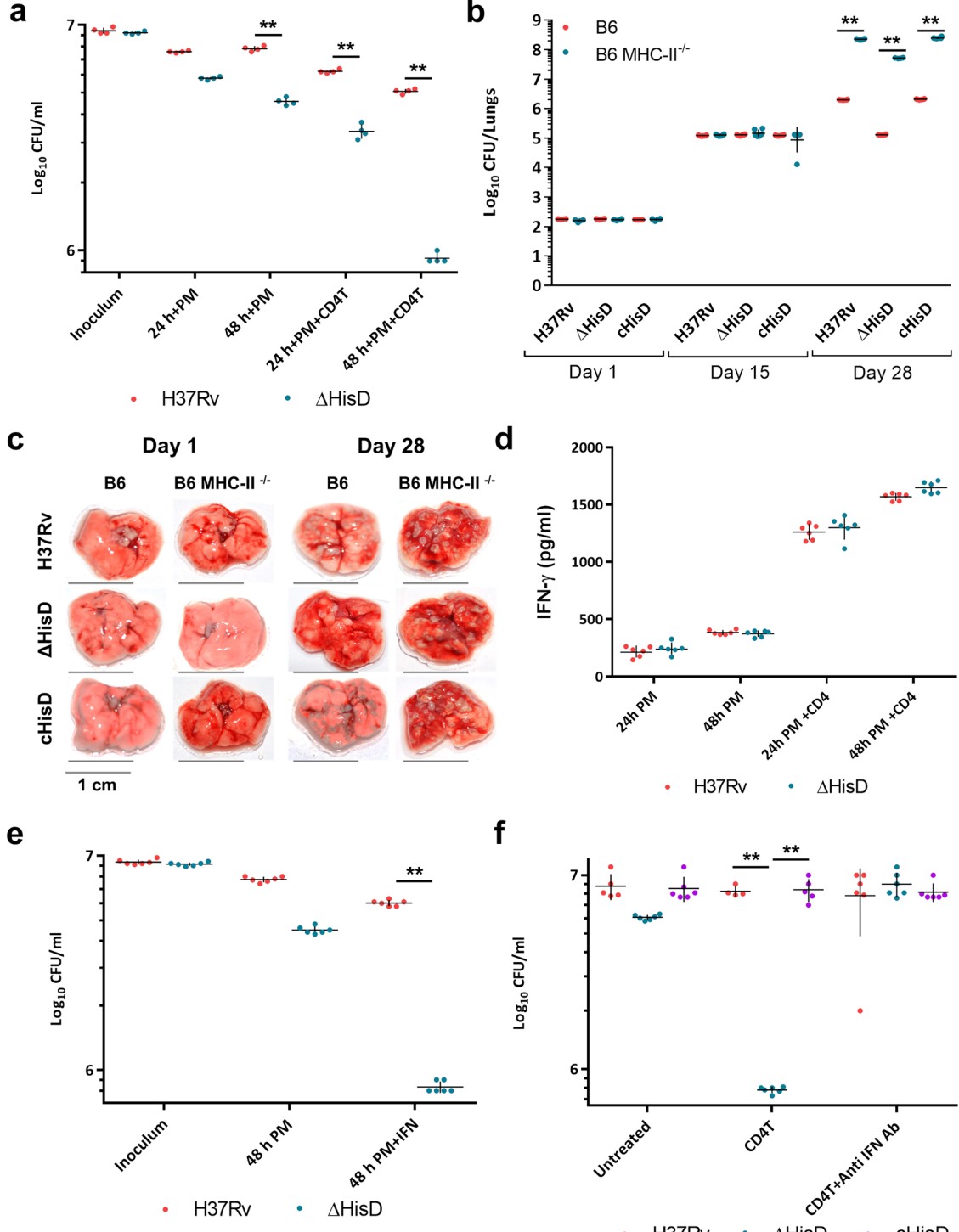

**Fig. 4 A CD4 directed IFN-γ mediated bacillary clearance of *Mtb* histidine auxotroph. a** Bacillary counts for Δ*hisD* are decreased by 10-fold compared to H37Rv at 48 h post-infection, suggesting a possible role of CD4 T-cell-based activation of macrophages in the enhanced clearance of Δ*hisD* ($n = 3$ independently acquired and cultured primary cells; mean and SEM; **P-value < 0.005). **b** The bacillary counts for H37Rv and c*hisD* each sharply rise as high as $10^8$ and for Δ*hisD* as $10^{7.5}$ on day 28 post infection in infected B6 MHC-II$^{-/-}$ mice lungs. ($n = 6$ individual mice per time point; mean and SEM; **P-value < 0.005). **c** A high degree of lung damage in the external pathophysiology is observed 28 days post infection with H37Rv, Δ*hisD* and c*hisD* in B6 MHC-II$^{-/-}$ null mice (pictures are representative of three individual mice lungs samples). **d** While un-stimulated macrophages show low titres of secreted IFN-γ, CD4 T cell activated macrophages show high levels of IFN-γ (1600 pg ml$^{-1}$) in culture medium 48 h post infection ($n = 6$ measurements from independently acquired and cultured primary cells; mean & SEM). **e** External supplementation of IFN-γ (~3000 pg ml$^{-1}$) post infection onto infected primary macrophages led to equivalent decrease in bacterial counts of Δ*hisD* as in case of CD4 T cell activated macrophages ($n = 6$ independently acquired and cultured primary cells; mean and SEM, **P-value < 0.005). **f** Abrogation of the bactericidal activity post treatment with anti-IFN antibodies observed in CD4 T cell stimulated macrophages infected with Δ*hisD* ($n = 6$ independently acquired and cultured primary cells; mean and SEM; **P-value < 0.005).

3000 pg ml$^{-1}$ of IFN-γ 3 h post infection. A decrease in the bacillary load of ΔhisD, equivalent to the CD4 T-cell activated primary macrophages was observed 48 h post infection (Fig. 4e and Supplementary Fig. 15a). Importantly, the anti-mycobacterial effect on ΔhisD was completely reversed upon incubation of CD4 T-cell activated primary macrophages with anti-IFN-γ antibodies (Fig. 4f), suggesting a direct role of IFN-γ in clearing the ΔhisD infection. However, CD4 T-cell stimulation and IFN-γ secretion are known to regulate a range of immune and metabolic processes and often cause off-target effects[72]. In vitro attenuated strains are prone to severe reduction in growth during in vivo and ex vivo infection conditions. To further ascertain the specificity of this anti-tubercular effect of CD4 T-cell-mediated IFN-γ against the ΔhisD, macrophages were infected with another attenuated strain of H37Rv (msh1 knockdown (Mycobacterial Secreted Hydrolase 1))[73]. Infected macrophages were stimulated with CD4 T cells and then treated with anti-IFN-γ antibodies. While msh1 knockdown exhibited mild attenuation in un-stimulated macrophages, CD4 T cell stimulation greatly reduced its growth independent of IFN-γ over 48 h (Supplementary Fig. 15b). Collectively, these results indicate a CD4 T-cell- mediated (though not limited to) clearance of ΔhisD, via an IFN-γ specific response.

To further validate the specificity of an IFN-γ mediated reductions in ΔhisD bacillary counts, C57BL/6-IFN-γ knockout mice (B6 IFN-γ$^{-/-}$) were infected with H37Rv, chisD and ΔhisD, along with a control infection in B6 mice. The infected lungs of B6 mice displayed a 10-fold decrease in bacillary counts of ΔhisD compared to H37Rv on day 28 post infection. However, infected lungs of B6 IFN-γ$^{-/-}$ displayed nearly identical bacillary loads for all the three strains (Fig. 5a). Analysis of the gross- and histopathology of the lungs showed that the H37Rv and chisD exhibited increased number of lesions and decreased air spaces on day 28 as compared ΔhisD-infected B6 mice lungs (Fig. 5b and Supplementary Fig. 16). However, the three strains displayed a similar degree of lung damage in B6 IFN-γ$^{-/-}$ mice (Fig. 5b and Supplementary Fig. 16). These data further assert the specificity of IFN-γ as the primary effector cytokine causing the clearance of ΔhisD. Of note, negligible expression of Mtb histidine biosynthesis and host histidine catabolism enzymes were observed in whole lung homogenates of H37Rv-infected B6 IFN-γ$^{-/-}$ mice (Fig. 5c). Furthermore, we examined the levels of HAL and HDC in H37Rv-infected primary murine macrophages either stimulated by CD4 T cells or CD8 T cells or supplemented with external IFN- γ. There was a significant rise in the expression levels of both HAL and HDC 48 h post infection, in macrophages activated with CD4 T cells (Supplementary Fig. 17, left panel) or treated with external IFN-γ (Supplementary Fig. 17, right panel). However, CD8 T-cell activation had inconsequential effects on the levels of histidine catabolizing enzymes (Supplementary Fig. 17, middle panel). To further authenticate the anti-mycobacterial effects of HAL and HDC up-regulation, infected macrophages activated with CD4 T cells were treated with specific inhibitors of HAL and HDC (inhibitor set 1: 1 mM L-1-methylhistidine, 5-methyl-1-H-imidazole and glycine (1:1:1 v/v) against HAL; and 2 mM histidine methyl ester against HDC. Inhibitor set 2: 50 mM L-1-methylhistidine, 5-methyl-1-H-imidazole and glycine (1:1:1 v/v) against HAL; and 100 mM histidine methyl ester against HDC. Inhibitor set 3: 400 µM L-histidine hydroxamate against HAL and HDC. Inhibitor set 4: 800 µM D-histidine against HAL and HDC)[74–76]. This led to a significant rise in the bacterial counts of ΔhisD 48 h post infection, reversing the effects of CD4 stimulation (Fig. 5d). These results substantially prove that the up-regulation of HAL and HDC is primarily associated with starving Mtb of free histidine.

**The levels of free histidine in the host cellular pool decreases over the course of infection.** In order to examine the effects of HAL and HDC up-regulation on the levels of free histidine, we quantified and compared the levels of free histidine in infected lungs on days 1, 15, 21 and 28 post infections. The exact levels of histidine were quantitated by a liquid-chromatography tandem mass spectrometry technique. An selected reaction monitoring (SRM) method was developed and found to be highly reproducible (Supplementary Fig. 18a). This method was then used to generate a linearity curve corresponding to a range of 1–250 ng ml$^{-1}$ of free histidine (Supplementary Fig. 18b), and its reliability was assayed through a reverse quality control analysis (Supplementary Fig. 18c). Quantitation of free histidine in infected lungs showed that the levels of free histidine remain unaltered on days 1 and 15. However, a sharp decline was observed on day 21 followed by a minimal rise on day 28 for B6 mice lungs samples infected with both H37Rv (orange line) and ΔhisD (blue line) (Fig. 6a and Supplementary Fig. 19a, b). The actual amount of free histidine for day 21 lung samples remained undetected, thus were assumed to be at par with the limit of detection for presentation purpose. Notably, the levels of free histidine in the B6 IFN-γ$^{-/-}$ mice lung samples infected with H37Rv were maintained at all the time points with negligible variation (Fig. 6a, green line and Supplementary Fig. 19c). This suggests that in the absence of IFN-γ, there is no evident depletion in the levels of histidine following Mtb infection.

To further correlate the decline in levels of free histidine with HAL and HDC, the levels of urocanate and histamine in lungs of B6 (orange line) and B6 IFN-γ$^{-/-}$ (green line) mice infected with H37Rv were estimated by semi-quantitative mass spectrometry. The results show a complete absence of urocanate on day 1 in infected B6 mice lungs and a relative rise on days 15, 21 and 28 was observed, suggesting an accumulation of the histidine catabolism by-product as the infection progressed (Fig. 6b and Supplementary Fig. 20a). Remarkably, the levels of urocanate remain completely undetected in B6 IFN-γ$^{-/-}$ mice lungs throughout the infection (Fig. 6b); corroborating with the absence of HAL expression (Fig. 5c). A similar rise in the abundance of histamine was detected in the B6 mice lungs over the course of 28 days of infection (Fig. 6c and Supplementary Fig. 20b). Interestingly, in B6 IFN-γ$^{-/-}$ mice lungs, a significant abundance of histamine was detected on day 15 which was followed by sharp decline on days 21 and 28 (Fig. 6c). This could possibly be attributed to the innate immune response that is active primarily in the early phases of infection. This innate immune response is characterised by a rapid influx of macrophages mediated by histamine amongst many immune modulators. Of note, the levels of methyl-histidine, a known excretory by-product synthesised via a post-translational methylation of histidine[77,78], were also explored. The results indicated a rise in the abundance of methyl-histidine in infected B6 mice lungs, whereas its levels in infected B6 IFN-γ$^{-/- g}$ mice lungs showed a sharp decline on days 21 and 28 (Fig. 6d and Supplementary Fig. 20c).

As a proof of concept, we also examined the levels of intracellular free tryptophan (simplified as "free tryptophan") in the lung homogenates of H37Rv-infected B6 and B6 IFN-γ$^{-/-}$ mice. An earlier study shows that following infection, IFN-γ mediates the up-regulation of the tryptophan catabolising enzyme Ido1[6]. An SRM method was developed and the linearity curve was generated for a range of 1–100 ng ml$^{-1}$ and was subjected to quality control analysis (Supplementary Fig. 21a, b). The results clearly showed a decline in free tryptophan levels post day 21 of infection, specifically in B6 mice (Fig. 6e and Supplementary Fig. 21c, d). Together these findings reveal that HAL and HDC deplete the levels of free histidine within the cellular environment following Mtb infection. This further corroborates to the

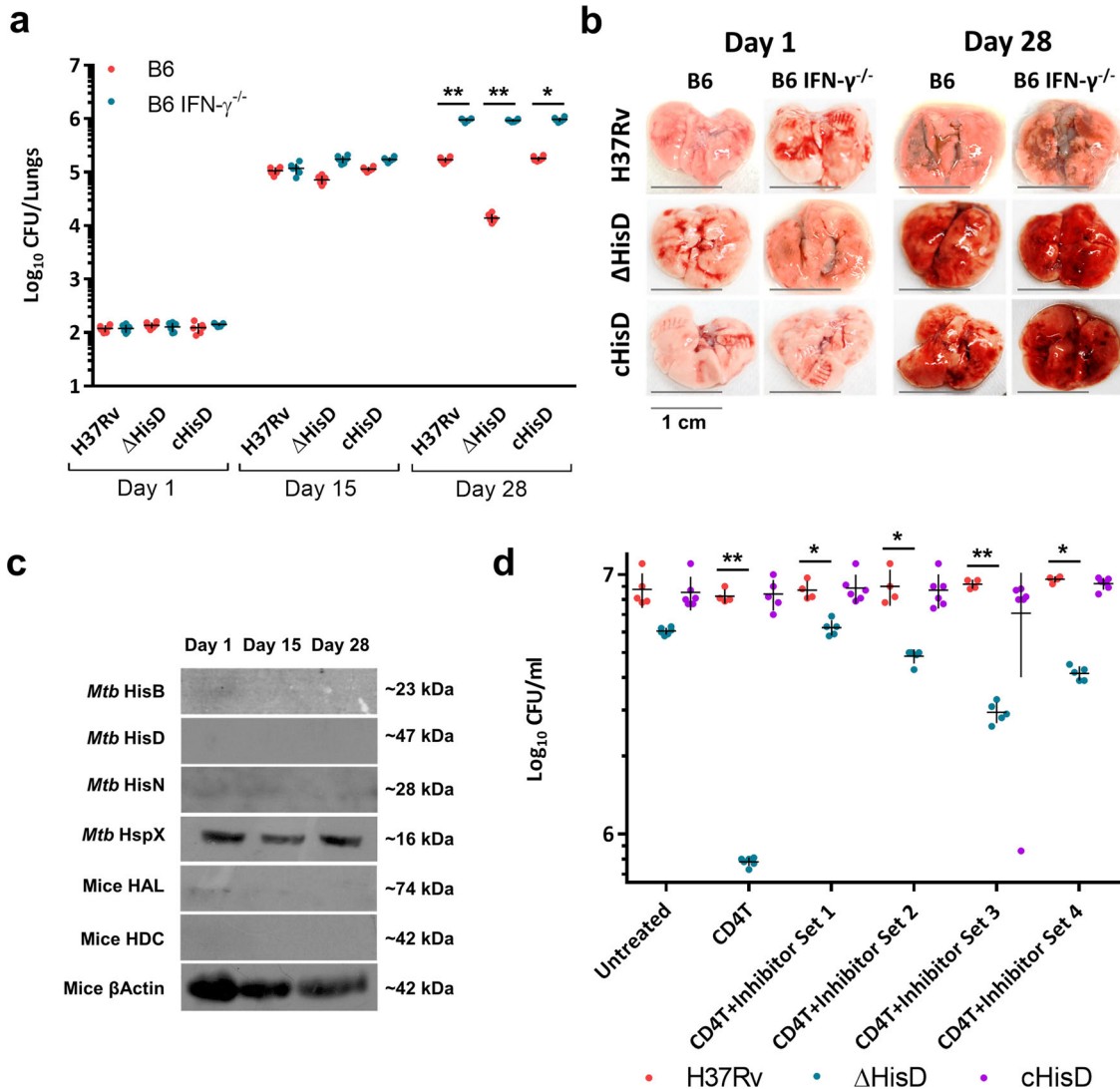

**Fig. 5 IFN-γ mediated control of HAL and HDC up-regulation. a** A 10-fold decrease in bacillary counts for Δ*hisD* compared to that of H37Rv and c*hisD* was detected in infected B6 mice lungs on day 28. In B6 IFN-γ$^{-/-}$ mice however, Δ*hisD*, c*hisD* and H37Rv exhibit near identical bacillary counts on day 28 (n = 6 individual mice per time point; mean and SEM; *P-value < 0.05, **P-value < 0.005). **b** A similar extent of lung damage was observed in B6 IFN-γ$^{-/-}$ mice infected with H37Rv, Δ*hisD* and c*hisD* on day 28 (pictures are representative of six individual mice lungs samples). **c** Negligible expression of host histidine sequestering enzymes and *Mtb* histidine biosynthesis enzymes in the whole lung lysates from B6 IFN-γ$^{-/-}$ mice infected with H37Rv. Mice β-Actin serves as the loading control. (Immunoblots are representative of samples obtained from three mice each time point). **d** Addition of specific inhibitors of HAL and HDC Δ*hisD*-infected primary macrophages significantly reverses the bactericidal effects of CD4 T cell activation, 48 h post infection (n = 5 independently acquired and cultured primary cells; mean and SEM; **P-value < 0.005).

corresponding rise in the levels of histidine catabolites and excretory metabolites, specifically in infected B6 mice lungs. Conversely, in the absence of IFN-γ$^{-/-}$ and the histidine catabolism thereafter, there is no evident change observed in the free histidine levels within the host cellular environment. This further establishes the mediation of IFN-γ in the homeostasis of free histidine levels in lung tissues during an *Mtb* infection.

## Discussion

Pervasive resistance to the first line anti-TB drugs poses a major threat to TB treatment[79], underscoring the need to develop new ways to curb the dissemination of *Mtb*. Amino acids are one of the fundamental requirements for survival of every organism. In addition to the canonical proteinogenic functions, amino acids such as histidine play key roles in the basic cellular metabolism as discussed above. Notably, many amino acids are involved in various immune functions and are actively involved in the

host–pathogen crosstalk. It was shown that in an on-going *Streptococcus* infection, the host and the bacteria compete for the intracellular arginine[80]. While the macrophages use arginine to generate reactive oxygen species to enhance bacterial clearance, the pathogen uses the arginine to avoid the phago-lysosomal fusion and subsequent acid stress. The amino acids asparagine is known to provide resistance to the invading *Mtb* against lysosomal acidification[81,82]. IFN-γ induced tryptophan metabolism not only restricts the availability of the amino acid for bacterial utilisation, but also generates kynurenine as a by-product. At the onset of the HIV infection, kyunurenine is known to bind the aryl hydrocarbon receptor (AhR) and induce the generation of Tregs that limit inflammation[83–85]. While it remains largely unexplored during an infection scenario, the histidine metabolism end product urocanate has been shown to induce the anti-inflammatory responses and suppression of delayed type hypersensitivity[86–88]. This suggests the possibility of

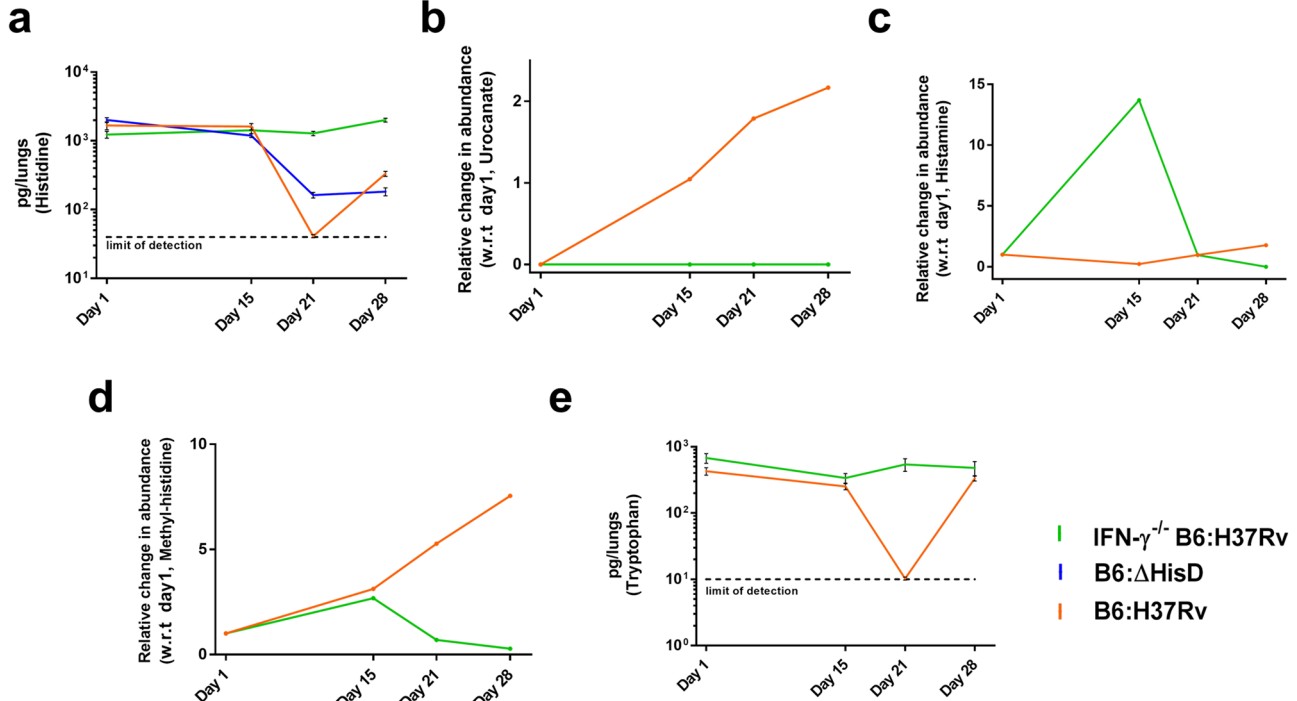

**Fig. 6 Depletion of free histidine levels in the host cellular milieu post infection. a** The levels of free histidine in B6 mice lungs infected with H37Rv and ΔhisD decline post 15 days of infection while these remain unchanged in infected B6 IFN-γ$^{-/-}$ mice lungs ($n = 5$ measurements from independently acquired and processed lung samples; mean and SEM). **b** While the relative abundance of urocanate increases post infection in B6 mice lungs infected with H37Rv, it remains undetected in infected B6 IFN-γ$^{-/-}$ mice lungs ($n = 5$ measurements from independently acquired and processed lung samples; mean only). **c** The levels of histamine in B6 mice lungs infected with H37Rv rise over the 4 weeks of infection, while there is a sharp decline in infected B6 IFN-γ$^{-/-}$ mice lungs post 15 days of infection ($n = 5$ measurements from independently acquired and processed lung samples; mean only). **d** The post transnationally modified excretory by-product methyl-histidine accumulates over the course of infection in B6 mice lungs infected with H37Rv while its abundance in infected B6 IFN-γ$^{-/-}$ mice lungs declines post 15 days of infection ($n = 5$ measurements from independently acquired and processed lung samples; mean only). **e** The levels of intracellular free tryptophan in B6 mice lungs sharply reduce 15 days post infection with H37Rv, while remaining nearly similar in infected B6 IFN-γ$^{-/-}$ mice lungs ($n = 5$ measurements from independently acquired and processed lung samples; mean and SEM).

the secondary immune-modulatory effects of histidine catabolism in addition to the primary sequestration effects induced by IFN-γ.

Unlike few other intracellular pathogens, *Mtb* genome contains genes encoding the enzymes for the biosynthesis of all the 20 amino acids[7]. If the invading bacillus can import free histidine from the host cell in sufficient quantities for fulfilling its physiological requirements, the histidine biosynthesis enzymes would be rather dispensable. It is worth mentioning that *Mtb* completely lacks the histidine utilisation (Hut) system, which is present in many species of eubacteria. The Hut system is comprised of genes that encode for both histidine catabolising enzymes (Supplementary Fig. 1a) and membrane associated histidine transporters (ABC transporters and permeases)[89]. Amongst the known species within the *Mycobacterium* genus, only *M. smegmatis* is known to possess a functional Hut system[90]. The *Mtb* H37Rv genome encodes for a plethora of membrane transporters such as: Rv2320c (cationic amino acids), Rv0488 (lysine), Rv2564 (glutamine), Rv1280c (oligopeptide) and Rv1979c, which are capable of transporting amino acids[91]. However, no specific histidine transport mechanism(s) has been identified in *Mtb*. Our study shows that as *Mtb* infects alveolar macrophages, a CD4 T-cell-mediated mechanism results in an enhanced production of IFN-γ. Subsequently, the IFN-γ downstream signalling upregulates the histidine catabolising enzymes HAL and HDC. This depletes the amount of free histidine in the host cellular milieu, eventually restricting its availability to *Mtb*. While the wild-type H37Rv and *chisD* counter this host immune response by utilising their inherent de novo histidine biosynthesis machinery and

continue to proliferate, bacilli lacking *hisD* are deprived of histidine and cleared during infection (Fig. 7).

Several genes encoding for the enzymes from the histidine biosynthesis pathway were shown to be essential for the survival of *Mtb* in a genome-wide mutagenesis study[11]. The Tropical Disease Research drug targets prediction results[24] collectively suggest that the enzymes of *Mtb* histidine pathway with no structural and functional homologues in humans, are novel anti-mycobacterial drug targets. Our study shows that the histidine biosynthesis pathway of *Mtb* is an integral part of its repertoire of the host immune evasion tools. Of note, this also indicates that *Mtb* has to continuously maintain an active histidine biosynthesis in order to persist within the host microenvironment following the onset of the adaptive immune responses. This renders the pathway highly vulnerable to external interventions such as small molecule inhibitors, which if efficiently disrupt the pathway, could possibly lead to rapid bacillary clearance during infection. In addition, this study also reveals a hitherto unknown facet of the host immunity targeted at restricting the access of *Mtb* to vital cellular resources and adds to the current understanding of host-*Mtb* interactions in the context of *Mtb* histidine metabolism.

## Methods
**Study design.** This study aims at elucidating the importance of histidine biosynthesis for the growth and survival of *Mtb* during the course of its infection and any possible host–pathogen crosstalk with reference to histidine as a critical resource. All the experiments were carried out a minimum of three times with independently grown/raised bacteria, cell line and mice. The sample sizes for the experiments presented in this study were estimated following the ARRIVE

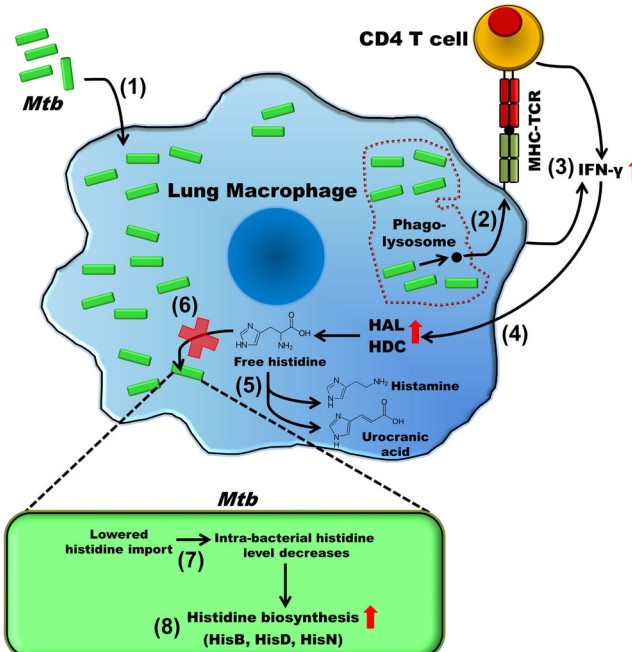

**Fig. 7 A schematic representation depicting the CD4 T-cell-directed and IFN-γ-mediated host immune response and the counter response by *Mtb*.** As the *Mtb* infects the lung macrophages (1), the bacilli are processed in the phago-lysosomes and the epitopes (black circle) are presented to the CD4 T cells (2). This results in an up-regulation of IFN-γ (3), which in-turn upregulates histidine catabolizing enzymes HAL and HDC (4). HAL and HDC subsequently catabolize free cellular histidine (5), thus restricting the availability of free histidine for bacilli to import (6). The reduction in exogenous histidine and its import diminish the intra-bacterial histidine level (7), thus inducing the de novo histidine biosynthesis in *Mtb* (8).

guidelines and using the recommendations of an earlier study[92]. For bacterial cultures a minimum of nine independently grown cultures were used. For ex vivo infection, a minimum of three independently acquired primary cells (isolated six individual mice) and/or cultured cell lines (three independent stocks) infected with a single bacterial strain at each time point were used. For infection in C57BL/6, C57BL/6-MHC-II$^{-/-}$ and C57BL/6-IFN-γ$^{-/-}$ mouse, a minimum of six mice were used per infection strain per time point. For immunoblotting, histopathology, immunohistopathology, ELISA and mass spectrometry analyses, a minimum of six lung lysate samples each extracted from an individual mouse were used. For RNASeq analysis three whole lungs samples for each condition, uninfected and infected were used, each extracted from an individual mouse. All mice used were female and ~5 weeks old. The number of biological replicates for each experimental set up is mentioned in the corresponding figure legends.

Experimental protocols for the animal experiments were approved by the Institutional Animal Ethics Committee (IAEC) (the permit numbers, NII/IAEC#321/13, NII/IAEC#439/17 and ICGEB/IAEC/2017/01/TACF-NII-11) upon the recommendations and standards prescribed by the Committee for the Purpose of Control and Supervision of Experiments on Animals (CPCSEA), Government of India.

All the resources and reagents utilised in this study are presented in Supplementary Table 6.

**Bacterial and cell culture maintenance**. The *Mtb* strains used in this study were maintained in Difco Middlebrook 7H9 broth liquid medium enriched with 10% ADC (albumin–dextrose complex). 0.05% Tween 80 and 0.2% glycerol at 37 °C with constant shaking at 180 rpm. 7H10 agar medium supplemented with oleic acid albumin–dextrose catalase (OADC) was used for solid medium cultures. For Δ*hisD*, both liquid and solid media were supplemented with 40 µg ml$^{-1}$ of histidine. H37Rv *msh1* knockdown strain was maintained as mentioned above and the gene knock down was controlled using anhydro-tetracycline as mentioned in an earlier report[73]. The *msh1* knockdown strain was generated by Dr. Nisheeth Agarwal from THSTI, India. *Escherichia coli* DH5α was grown in Luria–Bertani (LB) broth at 37 °C with constant shaking at 180 rpm for 12 h. The *M. smegmatis* mc$^2$-4517 was received from Dr. William Jacobs Jr., Albert Einstein College of Medicine, NY, USA. The *Mtb* strain H37Rv was obtained from BEI Resourses. The *Mtb* strain Δ*hisD* was received from Dr. Tanya Parish, University of Washington, USA. The *Mtb* Δ*hisD* strain is an unmarked deletion mutant for the ORF Rv1599

coding for the protein HisD. In brief, as mentioned in the original report, the procured Δ*hisD* strain was generated by a single crossover homologous recombination at the 3′ end of the ORF and the deletion was confirmed by southern blot analysis[18,19]. The *Mtb* strain c*hisD* was generated in house using a protocol described in the subsequent section. All the *Mtb* strains were analysed for the presence of contaminants, if any, by replica plating on 7H10 agar medium as mentioned above. The medium was supplemented with antibiotic cocktail PANTA and/or respective antibiotics, if any, prior to each experimental set up. All dilutions following infection and growth kinetics experiments were plated on 7H10 agar medium supplemented with antibiotic cocktail PANTA.

Monocytic cell lines RAW264.7 and THP1 were obtained from ATCC and maintained in RPMI 1640 medium supplemented with 200 mM glutamine, 100 µM of penicillin-streptomycin, 50 µM neomycin and 10% fetal bovine serum (at 37 °C in 5% CO$_2$). For infection experiments, cells were grown in RPMI 1640 medium without antibiotics for a minimum of 24 h prior to infection. Mice monocyte derived macrophages were isolated from buffy coats of blood extracted from mice infected with H37Rv, 15 days post infection. Cells were separated using Ficoll reagent, and macrophages were isolated by growing in GM-CSF (20 ng ml$^{-1}$) supplemented RPMI 1640 medium for 4 days before use. CD4 and CD8 T cells were magnetically isolated using a Positive Selection Kit from the aforementioned buffy coat. Naïve CD4 T cells were isolated similarly from uninfected mice. The primary macrophages and T cells were maintained in a medium similar to the cell lines and were supplemented with 15% fetal bovine serum.

**Complement strain generation**. The ORF coding for *hisD* – Rv1599 was PCR amplified using *Mtb* H37Rv genomic DNA as the template and gene specific forward primer (5′- CACCCATATGGTGCTTACCCGTATCGACTT -3′) and reverse primers (5′- TATAAGCTTTCATCATCGCTCGAACCTCCGCC -3′). The linear amplicon was then purified and cloned into an entry vector pENTR-D-TOPO as per manufacturer's manual and the entry clone was transformed into DH5α. The plasmid containing the gene of interest was isolated from the positive colonies and was digested at NdeI and HindIII sites. Finally, the digested fragment was further sub-cloned into the episomal expression vector pNIT1, yielding the expression construct, which was verified by nucleotide sequencing. The expression vector containing the target gene was transformed into Δ*hisD* by electroporation at 2500 V, 1000 Ω, 25 µF using 2 mm diameter electroporation cuvettes. The cells were then plated on a 7H10 agar medium supplemented with OADC and kanamycin (25 µg ml$^{-1}$) for the selection of transformed cells. Colonies appeared after 10–14 days, one of which was revived in a 10-ml 7H9 medium supplemented with ADC and kanamycin for expansion. The expression of recombinant gene was induced by the addition of 0.5 µM isovaleronitrile at an $A_{600}$ of 0.8. The clones were further confirmed by replica plating on 7H10 agar medium with/without external histidine supplementation and by immunoblotting analysis for the presence of HisD protein in the bacterial lysates.

**Protein overexpression and purification**. Expression clones of *hisB*[10] and *hisN*[25] were available, whereas *hisD* expression plasmid was prepared. Briefly, the corresponding ORF was amplified using gene specific primers and cloned into an expression vector pYUB-1062, overexpressed in a *M. smegmatis* mc$^2$-4517 system and the protein was purified to homogeneity by using Ni$^{2+}$-NTA affinity chromatography, followed by size exclusion chromatography. Briefly, the Rv1599 gene was PCR amplified using *Mtb* H37Rv genomic DNA as the template with specific primers with hexa-histidine tag at 5′ end of the forward primer (forward primer-5′-CACCCATATGCACCATCATCATCATCACGTGCTTACCCGTATCGACTT-3′ and reverse primer- 5′-TATAAGCTTTCATCATCGCTCGAACCTCCGCC-3′). The linear amplicons were purified and cloned into an entry vector pENTR-D-TOPO as per manufacturer's manual and the entry clone was transformed into DH5α. The plasmid containing the gene was isolated from the positive colonies and the plasmid was digested at NdeI and HindIII sites. Finally, the digested fragments were further sub-cloned into the shuttle expression vector pYUB-1062, yielding an expression construct, which was verified by nucleotide sequencing.

The overexpression and purification of these enzymes were carried out in a similar manner. The expression vector containing the target gene was transformed into *M. smegmatis* mc$^2$-4517 by electroporation at 2500 V, 1000 Ω, 25 µF using 2 mm diameter electroporation cuvettes. The cells were then plated on a 7H10 agar plate supplemented with OADC, kanamycin (25 µg ml$^{-1}$) and hygromycin (50 µg ml$^{-1}$) for the selection of transformed cells. Colonies appeared after ~72 h, one of which was inoculated in a 10-ml 7H9 broth supplemented with ADC and aforementioned antibiotics for revival. The culture was grown for about 24 h at 37 °C at 180 rpm. A primary culture of 50 ml 7H9 broth was inoculated using the revival culture. The culture was incubated for 12 h at 37 °C with shaking till $A_{600}$ was ~0.7. A 2000-ml 7H9 broth secondary culture was inoculated using the primary culture, and induced with 0.2% acetamide at an $A_{600}$ of ~0.7. The culture was harvested at 10,000×g for 30 min, 24 h post induction. Bacterial pellet was resuspended in 20 mM Tris buffer (at a pH of 8), containing 100 mM NaCl and protease inhibitor cocktail, and was homogenised. Cells were lysed using a cell disruptor at 4 °C. Subsequently, the soluble fractions were harvested by centrifugation at 10,000×g. Each protein was purified by affinity and size exclusion chromatography techniques using AKTA Explorer FPLC system (GE Healthcare, USA). Every purification step was maintained at 4 °C. The cell lysate in Tris-NaCl

loading buffer (at a standardised pH) was applied onto a Ni-NTA His-TrapFF column and non-specifically bound proteins were washed out with the sample loading buffer containing 50 mM imidazole. The protein was eluted in sample loading buffer containing 300 mM imidazole. The eluted protein fractions were pooled and concentrated using centrifugal units. Each protein was further purified by size exclusion using a superdex 200 pg column (HisB and HisD) or a superdex 75 pg column (HisN) in an AKTA Explorer FPLC system. The purity of each protein was examined on SDS–PAGE and respective protein identity was confirmed by mass spectrometry (MASCOT analysis).

**Animal maintenance**. C57BL/6 mice, C57BL/6-MHC-II$^{-/-}$, C57BL/6-IFN-$\gamma^{-/-}$ mice and NZW rabbits were bred and maintained in house at the Experimental Animal facility, National Institute of Immunology (NII). For infection experiments, the mice were maintained at Tuberculosis Aerosol Challenge Facility (TACF) International Centre for Genetic Engineering and Biotechnology (ICGEB), New Delhi, INDIA from a week prior to infection till the completion of experiments.

**Growth Kinetics of *Mtb* strains**. The three *Mtb* strains H37Rv, Δ*hisD* and c*hisD* were revived from glycerol stock and/or single colony picked from 7H10 agar medium and inoculated into 10 ml of 7H9 broth medium as mentioned in earlier section. For Δ*hisD* culture, the liquid medium was supplemented with 40 μg ml$^{-1}$ of histidine. Upon reaching an $A_{600}$ of 0.8, 500 μl of the respective cultures were inoculated independently into 10 ml of fresh liquid medium with appropriate supplements and the cultures were grown to an $A_{600}$ of 0.6. For determining the growth kinetics, 10 ml of fresh liquid medium with appropriate supplements were inoculated from the aforementioned cultures normalising the final $A_{600}$ to 0.1. The cultures were maintained for 21 days and $A_{600}$ and bacillary counts on appropriate agar medium were determined at specific intervals. For Δ*hisD* strain, the growth kinetics was analysed for three different conditions – histidine supplementation from the 1st day of inoculation; histidine supplementation up to 7th day post inoculation followed by histidine starvation; and histidine starvation up to 7th day post inoculation followed by histidine supplementation.

**Ex vivo and in vivo infection**. $10^5$ Monocytic cells or primary macrophages (stimulated with 5 μg ml$^{-1}$ of LPS 12 h prior to infection) were laid down per well of 24-well tissue culture plates. Cells were washed with PBS followed by the addition of antibiotic-free RPMI 1640 medium. The cells were then infected with *Mtb* strains- H37Rv, Δ*hisD* and c*hisD* (at an $A_{600}$ of 0.6) at a MOI of 10. H37Rv *msh1* Knockdown strain was infected similarly. After 3 h, non-phagocytosed bacteria were removed by washing the cells three times with PBS, and adherent monolayers were replenished with antibiotic-free culture medium. For CD4 T cells co-cultures, the infected macrophages were overlaid with CD4 T cells post washing of non-phagocytosed bacteria 3 h post infection. For IFN-γ stimulation, 3000 pg ml$^{-1}$ of the same was added to the infected macrophages post washing of non-phagocytosed bacteria 3 h post infection. For quenching of IFN-γ, anti-IFN-γ antibodies were added 6 h post activation with CD4 T cells. For abrogation of HAL and HDC activities, the inhibitors were added 6 h post activation with CD4 T cells. All ex vivo infections were carried out in Biocontainment level 3 facilities, NII.

For mice infection experiments, the mycobacterial strains were grown till $A_{600}$ of 0.6. The cultures were centrifuged; the pellets were washed with phosphate buffer saline tween- PBST (at a of pH 7.5) and resuspended in PBST to an $A_{600}$ of 0.6. These samples were used to infect ~5-weeks-old female mice through an aerosolic route (using an Einstein Contained Aerosol Pulmonizer type equipped with an Aerosol exposure chamber) with ~200 bacilli per animal. Animal infections experiments were carried out at TACF, ICGEB.

**Sample preparation for colony forming units/ bacterial cell counts**. For ex vivo infection- at 24 h and 48 h post infection, medium was removed and monolayers were lysed with 0.05% SDS. Serial dilutions were plated on agar plates (Middlebrook 7H10, 10% OADC enrichment). For in vivo infections, mice were euthanized on days 1, 15, 21, 28 and 63 post infections. Lungs and spleen were harvested for each mouse and weighed. A section of the tissue was saved for microscopy. The remaining sections were re-weighed, and homogenised using a barrel homogenizer. Appropriate dilutions of the lysates were plated on 7H10 agar plates as mentioned above. Colony-forming units were counted after incubation of the plates at 37 °C for 21 days for both in vivo and ex vivo infections. For mice, the bacterial counts were normalised using the weights of lungs.

**Sample preparation for H&E Staining and immunohistochemical experiments**. The lung tissue samples were embedded into wax blocks. The wax blocks were subjected to microtomy to generate sections, which were then stained with Haematoxylin and Eosin dyes at a commercial facility. For immunohistochemical analysis, appropriate biotin tagged primary antibodies were applied. This was followed by the application of HRP (horse radish peroxidase) tagged avidin bead for detection. Colour was generated using TMB (3,3′,5,5′-tetramethylbenzidine) as the substrate for HRP. The tissue sections were viewed in a bright field microscope at ×10 magnification.

**Antibody generation**. Polyclonal antibodies against the HisB, HisD and HisN were raised at in-house animal facility using NZW strain rabbit aged 15 weeks following a previously reported method[73]. Briefly, 1 mg ml$^{-1}$ of purified protein was emulsified with adjuvant (Freund's incomplete adjuvant) in 1:1 ratio using glass syringe connected with a three-way stopcock. The injection was administered subcutaneously with maximum volume of 500 μg of protein per site, and the maximum number of injection sites was two. Four booster immunisations were carried out with intervals of 2 weeks to generate maximum serum antibody titre. Blood sample was collected through retro-orbital route prior to every subsequent booster dose to extract serum antibodies generated against the respective protein. Serum was prepared after incubating the blood sample at 37 °C for 5–10 min followed by centrifugation at 3500×g for 15 min at 4 °C. Anti-sera for each protein was collected and stored at −80 °C. The antibody specificity was determined using complete mycobacterial lysates by immunoblotting. The purified proteins HisB, HisD and HisN were used at different concentrations to determine the affinity of the respective polyclonal antibody.

**Immunoblotting and ELISA**. Bacteria, infected cells and infected homogenised whole lungs were lysed in a bead beater using zirconia beads. The lysate was subjected to centrifugation at 10,000×g for 10 min at 4 °C and the supernatant collected was used for further analysis. The samples were supplemented with a protease inhibitor cocktail to prevent degradation of samples. The samples were subjected to SDS–PAGE followed by immunoblotting analysis. In house generated antibodies were used at appropriate concentrations followed by HRP linked Anti-Rabbit IgG secondary antibody for detection and quantitation of respective proteins in various lysate samples. ELISA kit was used to determine the levels of various proteins as per manufacturer's protocol. ELISA plate was read at a wavelength of 400 nm using a Microplate reader and relative expression levels were calculated using the absorbance values. The amount of protein samples used for immunoblotting and ELISA were normalised by determining the whole protein concentration using the bicinchoninic acid assay. The final individual protein quantities per sample were determined by comparing with available standards developed with purified homogenous protein samples. In order to account for the post-translational modifications, antibodies against the phosphorylated forms of these signalling molecules and transcription factors were used in immunoblotting. β-actin served as the internal control for mice proteins and Groel2 served as the internal control[93,94] for *Mtb* proteins. For *Mtb* proteins, expression of HspX was used as a marker for infection. Ponceau staining was performed by incubating the nitrocellulose membrane for 1 min in staining solution (0.1 g of Ponceau S stain granules in 100 ml of 5% glacial acetic acid).

**Whole lung RNA sample preparation for RNASeq**. The whole lungs extracted from uninfected and H37Rv-infected mice ($n = 3$) were finely sliced into tiny fragments which were stored in TRIzol (ThermoFisher) and provided to a commercial facility (Nucleome Informatics, Hyderabad), for further processing and next-generation sequencing. Briefly, the whole RNA was extracted from each sample using manufacturer's protocol. The extracted RNA was subjected to quality control analysis using Nanodrop2000 and Agilent 2100 Bioanalyzer. The samples with optimum quality were used to generate a cDNA library. The library was then subjected to a second quality control analysis using ThermoFisher Qubit 2.0 and Agilent 2100 Bioanalyzer. The qualifying libraries were pooled according to the effective concentrations and expected data volume, and subsequently fed into the Illumina HiSeq 2500 sequencer. The original data obtained from the high throughput sequencing platforms were transformed to sequenced reads by base calling. Raw data were recorded in a FASTQ file that contains sequenced reads and corresponding sequencing quality information. The sequenced reads (raw reads) were then filtered off to obtain the clean reads by removing the low quality reads (reads with >10% undetermined bases and the adaptor reads). The sequences were then mapped onto the reference genome (B6 mice) by using the alignment tool HISAT. Subsequently, the gene expression levels were estimated by counting the number of reads that map onto a particular gene from the reference genome using the software FeatureCounts. Finally, the differentially expressing genes were fetched using the software DESeq2. DESeq2 normalises the read counts, estimates the $P$-values and adjusts the $P$-value using FDR.

**RNASeq data analysis**. The differential expression data obtained from total mRNA next generation sequencing were analysed using Cytoscape for the presence of pathways (KEGG, Reactome, WikiPathways) using respective plugins- KEGG-Parser, KEGGScape, Reactome FI and WikiPathways. The network analysis was performed using GeneMANIA and Diffany plugins. Gene Ontology analysis was performed using ClueGO plugin. Further details are presented in Supplementary Table 7.

**Histidine, Tryptophan and metabolites analysis using mass spectrometry**. The infected lungs were homogenised and subjected to mild sonication to ensure the complete lysis of all mammalian cells, without affecting the integrity of the *Mtb* cells. The lysate samples were centrifuged at 10,000×g for 10 min at 4 °C and the supernatant were collected and passed through a centrifugal filter unit with a membrane of 3 kDa cutoff. The flow through samples (containing all biomolecules

less than 3 kDa molecular size) were collected, lyophilised and provided to a commercial facility (VProteomics, New Delhi) for histidine, tryptophan and metabolites quantification. The lyophilised samples were resuspended in a solvent (methanol, acetonitrile, water; 5:3:2 v/v) and vigorously vortexed. This ensured the removal of residual proteins, peptides, salts and lipids. The vortexed samples were subjected to two rounds of centrifugation at 15,000×$g$ for 10 min at 4 °C and the supernatants were collected and subjected quantitative mass spectrometry using the Thermo Scientific TSQ Fortis nanoLC MS/MS system equipped with Thermo Trace finder and Thermo Xcalibur software. Briefly, a SRM method was developed with an Electrospray Ionisation (ESI) source in positive mode using mass spectrometry grade histidine and tryptophan at different concentrations in Buffer A (50 mM ammonium formate), Buffer B (0.1% Formic acid in Acetonitrile); diluent – acetonitrile and water (4:1 (v/v); with a flow rate of 5 µls$^{-1}$ and linearity plots were generated. Ion source parameters were optimised for the mobile phase composition. The conditions for the ion spray were: ion source type- H-ESI; spray voltage- static; positive ion- 5000 V; negative ion- 2500 V; ion transfer tube temp- 573 K; vaporizer temp- 623 K; start time- 0 s; end time- 300 s; cycle time- 0.5 s; Q1 resolution- 0.7 (FWHM); Q3 resolution- 0.7 (FWHM). The SRM indicated that a single precursor with an $m/z$ value of 156.312, resulted in four products: $m/z$ 55.899 (13.21 V); 83.113 (21.76 V); 93.101 (30.74 V); and 110.012 (14.1 V; collision energies for each in parentheses), for histidine. For tryptophan, a single precursor with an $m/z$ value of 205.03, resulted in three products with m/z values of 118.042 (26.65 V); 146.071 (17.97 V); and 188.071 (10.09 V; collision energies for each in parentheses). The method was subjected to reproducibility and quality control analyses. A linearity plot was generated for increasing concentration of histidine and tryptophan, which was further used as a reference to quantify the amounts of free histidine and tryptophan in the samples. Baselines were corrected manually using base to base noise calculation combined with edge flatting for 30 scan cycles. The tolerance limits were set to 0.01 and 0.05 for histidine and tryptophan, and signal to noise ratio cut-offs were set to 80 and 50 for histidine and tryptophan, respectively. Peaks were selected manually based on retention time's nearest peak criteria. Peak signal to noise ratio cut-offs were set to 30 and 10 for histidine and tryptophan, respectively. Peaks were integrated for calculation of area with consistent time parameter and $m/z$ scanning. For detecting the histidine catabolism metabolites, a Thermo Scientific Q Exactive quadrupole-Oribtrap MS equipped with Thermo Trace finder software was used. The intensities were normalised using the respective histidine concentration for the respective sample and the relative abundance with respect to day 1 was determined. Following quantitation, the values for histidine, tryptophan and the metabolites were normalised with respect to the weight of the lungs.

**Statistics and reproducibility**. The experimental data points were subjected to FDR analysis for each independent experimental condition to eliminate any false positives and further statistically analysed using unpaired two tailed Student's $t$ test. Bars in histogram plots represent standard error of the mean (sem). $P$-values are presented in figures and figures legends. Non-significant data points are presented without $P$-value markers. All the representative plots were generated using Graphpad Prism 7.0.

**Reporting summary**. Further information on research design is available in the Nature Research Reporting Summary linked to this article.

## Data availability
The RNAseq data have been deposited in the ArrayExpress database at EMBL-EBI (www. ebi.ac.uk/arrayexpress) under accession number "E-MTAB-8663" using the Annotare submission tool[95]. Unprocessed western blot images are presented in Supplementary Figs. 22–24. All source data underlying the graphs and charts shown in the main and supplementary figures are presented in Supplementary Data 4. The raw files for the mass spectrometry analyses are available on request from the corresponding author.

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

## Acknowledgements
We sincerely acknowledge Dr. Tanya Parish of the University of Washington for providing us the *hisD* gene unmarked deletion H37Rv mutant. We thank Prof. William R. Jacobs of Albert Einstein College of Medicine, NY, USA for providing us the *M. smegmatis* expression system and the pYUB-1062 expression vector. *Mtb* H37Rv genomic DNA, monoclonal anti-*Mtb* HspX and Groel2 antibodies were obtained from the Biodefense and Emerging Infections Research Resources, NIH. The authors sincerely thank Mrs. Shanta Sen and Dr. Nagarajan from NII, New Delhi for helping us respectively in recombinant protein identification by mass spectrometry and in raising antibodies. We thank Dr. Lakshyaveer Singh and Mr. Mahendra Singh of TACF, ICGEB for helping us with animal infection experiments. The authors thank Dr. Prafullakumar B. Tailor of NII for his valuable suggestions. This work was supported by the Indian Council of Medical Research (ICMR) (Grant Number- 58/18/2015-BMS); Department of Biotechnology (Grant Number- BT/PR25690/GET/119/142/2017) and core funding to NII and THSTI.

## Author contributions
A.D. and A.A. contributed equally to this study. A.D. and B.K.B. conceptualised the study. A.D. and A.A. performed the experiments. A.D., B.J. and D.K. provided the purified protein samples for antibody generation. N.A. constructed the *msh1* knockdown strain of H37Rv. A.D. and A.A. analysed the data. A.D., A.A. and B.K.B. wrote the manuscript. All the authors reviewed and approved the manuscript.

## Competing interests
The authors declare no competing interests.
