## [Peer Review File · Communications Biology]

Reviewers' comments:

Reviewer #1 (Remarks to the Author):

The paper by Dwivedy et al. is an interesting and expansive study that characterises the failure of the hisD knockout strain of *M. tuberculosis* to cause disease in a mouse model. The authors show that the ability of the mouse to control and eliminate the hisD knockout strain is dependent on an IFN γ -dependent depletion of host histidine. The core experiments presented in the paper seem sound, but a number of improvements are required before I could recommend the paper for publication.

Major issues:

1. The paper needs a diagram of the histidine biosynthetic and degradation pathways as an introductory figure, to enable the reader to understand the chemistry being discussed.
2. In Figure 1B, histidine supplementation is stopped at day 7 (green curve), but the CFU count decreases from day 5, despite the A600 still increasing between days 5 and 7 in Fig 1A. This doesn't make physical sense.
3. Line 224: where is the evidence that *M. tuberculosis* resides in the cytoplasm of infected cells? None is presented in Reference 45, and my understanding is that the bacterium is thought to reside primarily in the phagosomes of host cells, and does not escape into the cytoplasm as is seen for *Listeria monocytogenes*, for example.
4. In Figure 3B, the expression of STAT1 is markedly increased on day 28, whereas at the same time point, STAT3 is markedly decreased. Why is that?
5. In Figure 5D, L-1-methylhistidine, 5-methyl-1-H-imidazole and glycine are added as 'specific' inhibitors at a final concentration of 50 mM, and histidine methyl ester at 100 mM. These seem like very large doses, with the potential to cause non-specific effects. In Katona et al. (ref 49), the K_i of 5-methyl-1-H-imidazole and glycine for HAL is 1.72mM, but many inhibitors of HAL are described in that paper with K_i values in the 10-100 μ M range). Histidine methyl ester is described in Brand (ref 51) as 'not inhibitory' to HAL at 5mM, and again, a series of better inhibitors are described in that paper, e.g. D- α -hydrazinoimidazolyl-propionic acid, which has a $K_i = 75\mu$ M. The possibility exists then that this experiment may not be the result of specific inhibition of histidine breakdown, and would be more convincing if repeated with more potent inhibitors.
6. Figure S19 – how are the areas under the curves selected? It might be helpful to show absolute rather than relative intensities. Also show the trace for Day 21 for H37Rv.

-

Major general issues:

1. References need to be supplied for some general unsupported statements e.g. line 106 "well known for their essentiality", a reference is needed for hisB; "it is widely known that the adaptive immune responses activate in approximately 15-20 days" line 150-151; "earlier studies" line 223.

2. Totally absent is any discussion about what is known about systems for histidine importation in M. tuberculosis – this clearly has a strong bearing on the interpretation of the data presented.

3. Throughout the paper, the English language usage is not always clear, and would benefit from editing. Some example are listed in the ‘minor issues’ list, but there are others.

Minor issues:

1. The nomenclature of bacterial proteins should be checked – e.g. ‘HISD’ should be written as ‘HisD’.

2. Clarity of figures throughout could be improved. For example, the vertical legends are hard to read, and colours would be clearer than checkerboard hatching in bar graphs.

3. Line 235 – does a ‘conspicuous difference’ mean a ‘statistically significant difference’?

4. Line 238 – ‘cfus’ – cell counts

5. Line 242 – It is unclear what ‘log fold decline’ precisely means.

6. Line 246 – ‘the infected B6 mice lungs displayed increased number of lesions on day 28 for H37Rv and chisD infection as that of Δ hisD’ – this does not make sense.

7. Line 274 – It is unclear what ‘one log fold’ precisely means.

8. Line 278 – ‘lesser air spaces’ – Do the authors mean ‘fewer air spaces’ or ‘smaller air spaces’?

9. Line 283/286 – ‘expressions’ should be ‘expression’

10. Line 302 – define SRM. Shaun Lott, University of Auckland.

Reviewer #2 (Remarks to the Author):

The manuscript “De novo histidine biosynthesis protects Mycobacterium tuberculosis from host IFN- γ mediated histidine starvation” by Abhisek Dwivedy et al is an interesting and novel study exploring the interaction between host and pathogen metabolism, specifically histidine metabolism and immunology and therefore adds to a body of research in the emerging field of immunometabolism. As such this is an important study which advances the TB field. I have some concerns about the manuscript and data in its current form which need to be addressed.

1. Overall this manuscript needs editing as many of the sentences are very clunky and/or overly lengthy and some are difficult to understand e.g. P3 L37-40 and other examples below (the list is not exhaustive).

2. ALL figure legends also need to be significantly improved for clarity. E.g. Fig1a. This figure needs to be improved as it is difficult to understand and poorly presented. How many times were these experiments biologically replicated? (please state in the legends?). The colours are far too similar. It’s impossible to read the key for the graph sideways.

3. Amino acids as mediators of the cross-talk between host and pathogen is becoming established. That amino acids have a significant influence on immune responses of the host is also established. It would be useful if the authors briefly reviewed this in the intro.

4. I'm very unconvinced by all the Western blot results presented in the manuscript. These blots are a mess and in (Fig 3 and Fig 5) the loading controls are not even the same. Normalising to HspX seems a very poor control for bacterial load as Mtb alters expression of this protein during infection. Please justify this choice in the text including reference(s). Some of the data from the Westerns is not interpreted correctly in the manuscript text. For example not all of the HIS biosynthesis genes increase expression as the infection progresses.

5. I am more convinced by the ELISA data but this lacks error bars and data is only shown for selected days. This is particularly an issue for the murine genes where only small increases of expression are observed.

6. Overall I'm unconvinced of the value of the RNA-seq data in the context of this manuscript. RNA seq has been performed many times in this TB model. Signalling pathways are post-translationally regulated and therefore using a protein kinase array may have provided more convincing data.

7. Co-culture experiments are interesting but what happens if you use any other attenuated mutant of Mtb? The association between an in vivo attenuated Mtb mutant being more attenuated with an increased immune response isn't novel. Please discuss in the text.

8. The measurement of total histidine in the lungs is interesting but all the timing don't quite tally up and need to be clearly discussed. For some of the figures time-points have been excluded. Why? This figure also needs to be improved. As a control what happens to levels of another amino acid levels during the infection? Changes in amino acid levels is a well-established response to invasion by pathogens.

Minor comments

9. P3, L35 the genome wasn't invented it was sequenced

10. P4 L50-53 rewrite this whole sentence

11. P5L67 Replace coherence with concordance

12. P5, L70-72 Rewrite this sentence

13. Infection pervasiveness??? Rewrite

14. P7, L125: You write "Histidine is an abundantly available resource in the host cellular milieu, and is readily available for the bacillus for uptake, unless the host has its own mechanism to restrict histidine for example through efficient sequestration". What's the evidence for this? This sentence needs a reference? Intracellularly? Extracellularly??? Where in the host are you talking about?

15. P8, 128-133. Please rewrite. This doesn't quite make sense

(Authors' responses are present in bold font)

Reviewer #1 (Remarks to the Author):

The paper by Dwivedy et al. is an interesting and expansive study that characterises the failure of the hisD knockout strain of *M. tuberculosis* to cause disease in a mouse model. The authors show that the ability of the mouse to control and eliminate the hisD knockout strain is dependent on an IFN γ -dependent depletion of host histidine. The core experiments presented in the paper seem sound, but a number of improvements are required before I could recommend the paper for publication.

>> We thank you for the in-depth analysis of our study.

>> The manuscript was proof read by a senior colleague from our institute. The manuscript was then checked for further grammatical correction, vocabulary usage and readability using the softwares: ProWritingAid and Grammarly (basic version). In our view, the revised manuscript reads better.

Major issues:

1. The paper needs a diagram of the histidine biosynthetic and degradation pathways as an introductory figure, to enable the reader to understand the chemistry being discussed.

>> We have included a diagram (Fig. S1 A; Page- 3; Line- 51) summarizing the histidine biosynthesis and catabolism pathways in the revised manuscript.

2. In Figure 1B, histidine supplementation is stopped at day 7 (green curve), but the CFU count decreases from day 5, despite the A600 still increasing between days 5 and 7 in Fig 1A. This doesn't make physical sense.

>> We have repeated the growth kinetics experiment for the H37Rv, Δ hisD and chisD strains with the similar conditions as was presented in the original manuscript. The current results have been updated in the text (Page- 5; Line- 73-79) and in Fig. 1 A and B.

3. Line 224: where is the evidence that *M. tuberculosis* resides in the cytoplasm of infected cells? None is presented in Reference 45, and my understanding is that the bacterium is thought to reside primarily in the phagosomes of host cells, and does not escape into the cytoplasm as is seen for *Listeria monocytogenes*, for example.

>> We have cited few additional references (Reference no. 60, 61 & 62), citing studies that show the presence of *Mtb* in cytoplasm, in the revised manuscript.

>> These reports and/or reviews suggest that *Mtb* resides in phagosomal, phago-lysosomal and cytosolic components of the host cells. Also, cytosolic escape is one the

major immune evasion tools primarily utilized to avoid acidification and lysosomal mediated bacterial clearance.

4. In Figure 3B, the expression of STAT1 is markedly increased on day 28, whereas as the same time point, STAT3 is markedly decreased. Why is that?

>> STAT1 and STAT3 bind to the same receptor complex and exhibit competitive binding. During infection, STAT3 is activated by the innate immune responses, mainly via type I IFNs. On the other hand, STAT1 is activated through adaptive immune mechanism, primarily by IFN- γ . STAT1 controls many pro-inflammatory responses causing bacterial clearance and apoptosis. STAT3 is involved in both pro- and anti-inflammatory signalling cascades which are regulated by the effector cytokines (Reference no. 54, 55, 56, 57 and 58 of the revised manuscript).

>> We have concisely explained the significance of this STAT1-STAT3 dynamics in the revised manuscript (Page- 11; Line- 234-243).

5. In Figure 5D, L-1-methylhistidine, 5-methyl-1-H-imidazole and glycine are added as 'specific' inhibitors at a final concentration of 50 mM, and histidine methyl ester at 100 mM. These seem like very large doses, with the potential to cause non-specific effects. In Katona et al.(ref 49), the K_i of 5-methyl-1-H-imidazole and glycine for HAL is 1.72mM, but many inhibitors of HAL are described in that paper with K_i values in the 10-100 μ M range). Histidine methyl ester is described in Brand (ref 51) as 'not inhibitory' to HAL at 5mM, and again, a series of better inhibitors are described in that paper, e.g. D- α -hydrazinoimidazolyl-propionic acid, which has a $K_i = 75\mu$ M. The possibility exists then that this experiment may not be the result of specific inhibition of histidine breakdown, and would be more convincing if repeated with more potent inhibitors.

>> As suggested by the reviewer, we have repeated the small molecule mediated inhibition of HAL and HDC using four set of inhibitor:

Inhibitor Set 1: 1 mM L-1-methylhistidine, 5-methyl-1-H-imidazole and glycine (1:1:1 v/v) against HAL; and 2 mM histidine methyl ester against HDC.

Inhibitor Set 2: 50 mM L-1-methylhistidine, 5-methyl-1-H-imidazole and glycine (1:1:1 v/v) against HAL; and 100 mM histidine methyl ester against HDC.

Inhibitor Set 3: 400 μ M L-histidine hydroxamate against HAL and HDC.

Inhibitor Set 4: 800 μ M D-histidine against HAL and HDC

>> The results for the inhibitor Set 2 was presented in the original manuscript. The new experiments suggest that the same inhibitor set produced almost similar results at 1/50th of the original concentration (Inhibitor Set 1).

>> While we were unable to procure D- α -hydrazinoimidazolyl-propionic acid, the inhibitors L-histidine hydroxamate and D-histidine were readily available for purchase.

These inhibitors also exhibit K_i in the micromolar range as that of α -hydrazinoimidazolyl-proprionic acid.

>> The results are updated in the text (Page- 15; Line- 326-333) and in Fig. 5D of the revised manuscript.

6. Figure S19 – how are the areas under the curves selected? It might be helpful to show absolute rather than relative intensities. Also show the trace for Day 21 for H37Rv.

>> The desired curves were identified both manually (using the retention time profile, and the shape of the curves) and automatically by the Thermo Trace finder software (integration start point to stop point coupled with dynamic linking). On identification of the desired peak, the area under the curve was calculated using the aforementioned software.

>> All the intensities presented for histidine, tryptophan and the histidine metabolites are absolute. For histidine and tryptophan we have quantitated the exact amounts of the respective molecule in each sample. However, the metabolites' intensities were normalized using their respective histidine values and thus are presented as “relative change in abundance w.r.t day 1”.

>> We were unable to detect any histidine in B6 mice lungs infected with both H37Rv and $\Delta hisD$ on day 21 post infection. The blank traces for the aforementioned samples have been updated in the updated manuscript in the Fig. S19.

Major general issues:

1. References need to be supplied for some general unsupported statements e.g. line 106 “well known for their essentiality”, a reference is needed for hisB; “it is widely known that the adaptive immune responses activate in approximately 15-20 days” line 150-151; “earlier studies” line 223.

>> The study describing the essentiality of *Mtb* HisB has been cited in the revised manuscript (Reference no. 22).

>> The studies reporting the onset of adaptive immune response in a murine model of *Mtb* infection have been cited in the revised manuscript (Reference no. 27 & 28).

>> The study highlighting the critical role of CD4 T cells in providing protective immunity in a murine model of *Mtb* infection has been cited in the revised manuscript (Reference no. 59).

2. Totally absent is any discussion about what is known about systems for histidine importation in *M. tuberculosis* – this clearly has a strong bearing on the interpretation of the data presented.

>> Many bacterial species possess the “histidine utilization (Hut) system” for regulating the levels of cytosolic histidine. Many membrane proteins such as permeases and ABC

transporter are members of the Hut system that regulate histidine import. While *Mtb* completely lacks the Hut system, it possess a range of permeases and ABC transporters involved in import of cationic amino acids, glutamine etc. However no specific histidine import mechanism has yet been identified in *Mtb*.

>> The Hut system and its presence in the Mycobacterial family has been concisely described in the discussion section of the revised manuscript (Page- 19; Line- 413-422).

3. Throughout the paper, the English language usage is not always clear, and would benefit from editing. Some example are listed in the 'minor issues' list, but there are others.

>> The manuscript was proof read by a senior colleague from our institute. The manuscript was then checked for further grammatical correction, vocabulary usage and readability using the softwares: ProWritingAid and Grammarly (basic version).In our view, the revised manuscript reads better.

Minor issues:

1. The nomenclature of bacterial proteins should be checked – e.g. 'HISD' should be written as 'HisD'.

>> The nomenclature for the bacterial proteins has been updated in the revised manuscript.

2. Clarity of figures throughout could be improved. For example, the vertical legends are hard to read, and colours would be clearer than checkerboard hatching in bar graphs.

>> The vertical legends have been replaced with horizontal legends in the revised manuscript.

>> The histogram bar graph color scheme has been updated in the revised manuscript.

3. Line 235 – does a 'conspicuous difference' mean a 'statistically significant difference'?

>> "Conspicuous difference", here meant a noticeable change in the bacterial cfu count such as one log fold change. For this particular experiment (Page- 13; Line- 264-266), despite being statistically significant, the difference in the bacterial counts of $\Delta hisD$ in un-stimulated and CD 8 T cell activated macrophage is mere 0.2 log folds. Also, This phrase has been removed from the updated manuscript.

4. Line 238 – 'cfus' – cell counts

>> The term "cfus" or "cfu" has been replaced with "bacterial counts" or "bacillary counts" in the revised manuscript.

5. Line 242 – It is unclear what 'log fold decline' precisely means.

>> One "log fold decline" denotes a 10 times decrease. Similarly two "log fold decline" denotes a 100 times decrease.

6. Line 246 – ‘the infected B6 mice lungs displayed increased number of lesions on day 28 for H37Rv and *chisD* infection as that of Δ *hisD*’ – this does not make sense.

>> The sentence has been updated to “Lung tissue damage was significantly more pronounced in B6 mice infected with H37Rv and *chisD* as compared to Δ *hisD* on day 28 post infection. However, B6 MHC-II^{-/-} mice infected with H37Rv, Δ *hisD* and *chisD*, displayed near identical levels of lung damage (Figure 4C).” (Page- 13; Line- 272-275).

7. Line 274 – It is unclear what ‘one log fold’ precisely means.

>> “One log fold” denotes 10 times. Similarly “two log fold” denotes 100 times.

8. Line 278 – ‘lesser air spaces’ – Do the authors mean ‘fewer air spaces’ or ‘smaller air spaces’?

>> The phrase ‘lesser air spaces’ has been updated to “decreased air spaces” in the revised manuscript (Page- 15; Line- 313). This essentially denotes a combination of both “fewer air spaces” as well as “smaller air spaces”, as observed in an acute *Mtb* infection.

9. Line 283/286 – ‘expressions’ should be ‘expression’

>> This has been duly updated in the revised manuscript.

10. Line 302 – define SRM

>> SRM or “Selected Response Monitoring” has been defined in the updated manuscript (Page- 16; Line- 342).

(Authors' responses are present in bold font)

Reviewer #2 (Remarks to the Author):

The manuscript “De novo histidine biosynthesis protects Mycobacterium tuberculosis from host IFN- γ mediated histidine starvation” by Abhisek Dwivedy et al is an interesting and novel study exploring the interaction between host and pathogen metabolism, specifically histidine metabolism and immunology and therefore adds to a body of research in the emerging field of immunometabolism. As such this is an important study which advances the TB field. I have some concerns about the manuscript and data in its current form which need to be addressed.

>> We appreciate the reviewer's efforts in in-depth analysis of our study.

1. Overall this manuscript needs editing as many of the sentences are very clunky and/or overly lengthy and some are difficult to understand e.g. P3 L37-40 and other examples below (the list is not exhaustive).

>> The manuscript was proof read by a senior colleague from our institute. The manuscript was then checked for further grammatical correction, vocabulary usage and readability using the softwares: ProWritingAid and Grammarly (basic version). In our view, the revised manuscript reads better.

2. ALL figure legends also need to be significantly improved for clarity. E.g. Fig1a. This figure needs to be improved as it is difficult to understand and poorly presented. How many times were these experiments biologically replicated? (please state in the legends?). The colours are far too similar. It's impossible to read the key for the graph sideways.

>> The vertical legends in the figures have been replaced with horizontal legends in the revised manuscript.

>> The histogram bar graph colour scheme has been updated in the revised manuscript.

>> Every figure legend is presented with a phrase in brackets- “(n=12; mean and SEM; *P-value<0.05)”. Here, “n” represents the number of biological replicates, error bars represent the “SEM” and the “*” shown in the figure represents a statistical significance of less than 0.05.

>> Also the general details of the number of experiments, biological replicates and experimental groups are presented in the Materials and Methods, sub-sections “Study Design” (Page- 22; Line- 446-462) and “Statistical Analysis” (Page- 33; Line- 719-724).

3. Amino acids as mediators of the cross-talk between host and pathogen is becoming established. That amino acids have a significant influence on immune responses of the host is also established. It would be useful if the authors briefly reviewed this in the intro.

>> The importance of histidine in addition to its proteinogenic functions such as their roles in hormonal control, maintenance of cellular pH has been concisely described in the introduction section (Page- 3; Line- 45-51).

>> The importance of various amino acids in modulating the host pathogen interaction (acidification and acid resistance by asparagine) and their roles in regulating the host immune responses (for example, induction of Tregs by tryptophan) has been briefly described in the discussion section (Page- 19; Line- 393-409).

4. Im very unconvinced by all the Western blot results presented in the manuscript. These blots are a mess and in (Fig 3 and Fig 5) the loading controls are not even the same. Normalising to HspX seems a very poor control for bacterial load as *Mtb* alters expression of this protein during infection. Please justify this choice in the text including reference(s). Some of the data from the Westerns is not interpreted correctly in the manuscript text. For example not all of the HIS biosynthesis genes increase expression as the infection progresses.

>> We have repeated the western blotting in an attempt to generate cleaner results and a few have been duly updated in the manuscript (Fig. 2A).

>> We accept that *Mtb* HspX is not a loading control as the expression of this protein is known to vary greatly. We however had used HspX as a marker of infection (Page- 7; Line- 127). We have presented a cleaner blot for HspX in the revised manuscript (Fig. 2A).

>> In addition, we have also incorporated western blots for *Mtb* Groel2 (Fig.2 A and B), which is widely used as loading control for *Mtb* proteins (Reference no. 88 & 89), in the revised manuscript. Using the expression levels of Groel2, we normalized the sample amount with respective bacterial counts.

>> We have also incorporated Ponceau stained representative nitrocellulose membranes as an indicator of equal loading, in the revised manuscript (Fig.S3 A and C) (Page- 7; Line- 127 & 135).

>> The increase in the histidine biosynthesis enzymes other due to differential conformation of the protein. For example, the functional unit of HisB is a 24 mer complex possessing 24 active sites (Reference no. 10). On the other hand, HisN is a dimer possessing two active sites (Reference no. 23). The idea behind the statement is that we observe a rise in the expression levels of these enzymes 2-3 weeks post infection.

5. I am more convinced by the ELISA data but this lacks error bars and data is only shown for selected days. This is particularly an issue for the murine genes where only small increases of expression are observed.

>> The ELISA data presented as histograms in Fig. S3 and S12 have been updated with error bars in the revised manuscript.

6. Overall I'm unconvinced of the value of the RNA-seq data in the context of this manuscript. RNA seq has been performed many times in this TB model. Signalling pathways are post-translationally regulated and therefore using a protein kinase array may have provided more convincing data.

>> RNA seq and microarray studies have been performed numerous times in the murine TB model involving specific cell types or tissues. This study presents the transcriptome of the whole lungs infected with H37Rv in an attempt to understand the changes occurring at an organ/global level.

>> The interaction network predicted in this study is primarily populated with JAKs, STATs and IRFs. In the original manuscript we had presented the levels of phosphorylated STAT1 and STAT3 (Fig. 3B and Fig. S12), taking into account the post-translational modification of proteins (Kindly visit the “Antibodies, chemicals and reagents” section of the “Reagents, Resources and Tools” table of the original manuscript.)

>> In the revised manuscript, we have performed western blotting analysis for the phosphorylated forms of IRFs and JAK2 (Fig. 3B) (Page- 30; Line- 651-653). We have also determined the levels of IFN- γ and IL-6 receptors (Fig. 3B).

7. Co-culture experiments are interesting but what happens if you use any other attenuated mutant of Mtb? The association between an in vivo attenuated Mtb mutant being more attenuated with an increased immune response isn't novel. Please discuss in the text.

>> To address the query number 7 raised by Reviewer 2, we collaborated with a microbiologist, Dr. Nisheeth Agarwal, from THSTI, Faridabad, INDIA. A H37Rv knockdown strain for the gene *msh1* was generated during an earlier collaborative study (Reference no. 69). Using this strain, experiments were performed and the results were updated in the revised manuscript. We have included Dr. Nisheeth as a co-author in the revised manuscript.

>> >> In the revised manuscript we have presented the *ex vivo* infection profile of an H37Rv strain with a CRISPRi-mediated knockdown of the secretory hydrolase Msh1. While, the growth profile of Msh1 knockdown strain is similar to that of the wild type H37Rv in primary macrophages, it exhibits severe attenuation upon CD4 T cell stimulation of the infected macrophages. Interestingly, upon addition of anti-IFN- γ antibodies, following the CD4 activation of the infected macrophages, the *msh1* (Mycobacterial Secretory Hydrolase 1) knockdown strain remains attenuated. This suggests a minimal role of secreted IFN- γ or its downstream signalling pathways in intracellular survival of *msh1* knockdown strain. This is in stark contrast to the $\Delta hisD$ strain, where the attenuated intracellular growth phenotype is restored upon abrogation of secreted IFN- γ , clearly indicating a direct effect.

>> The results of these experiments are discussed in the text (Page- 14; Line- 295-304) and presented in Fig. S15 B of the updated manuscript.

8. The measurement of total histidine in the lungs is interesting but all the timing don't quite tally up and need to be clearly discussed. For some of the figures time-points have been excluded. Why? This figure also needs to be improved. As a control what happens to levels of another amino acid levels during the infection? Changes in amino acid levels is a well-established response to invasion by pathogens.

>> We were unable to detect any histidine in B6 mice lungs infected with both H37Rv and $\Delta hisD$ on day 21 post infection. The blank traces for the aforementioned samples have been updated in the updated manuscript in the Fig. S19.

>> In addition to histidine, we have also quantified the levels of free tryptophan in the lung lysates of B6 and B6- IFN- $\gamma^{-/-}$ mice infected with H37Rv. As described in an earlier study (Reference no. 6), in a murine model of *Mtb* infection, an IFN- γ mediated mechanism up-regulates the tryptophan catabolism enzyme IDO, intended to starve the bacilli of tryptophan. We confirm that, indeed on day 21 and day 28 post infection with *Mtb*, there a severely decline of free tryptophan in the lung lysates, specifically in the B6 mice, but not in the B6- IFN- $\gamma^{-/-}$ mice.

>> The results for the aforementioned experiments have been updated in the text (Page-17; Line- 375-381) and presented in Fig. 6E and Fig. S21.

Minor comments

9. P3, L35 the genome wasn't invented it was sequenced

>> The word presented in the line number 35 of the original manuscript is advent (Line- 38 in the revised manuscript).

10. P4 L50-53 rewrite this whole sentence

>> This statement has been rephrased in the revised manuscript as “The scenario however is interestingly different in an *in vivo* mice model of infection. We show that two weeks post infection, the host through an IFN- γ mediated mechanism up-regulates its histidine catabolizing enzymes- histidine ammonia-lyase (HAL) and histidine decarboxylase (HDC), possibly to starve the *Mtb* of free intracellular histidine. However, the wild-type *Mtb* are able to kick-start the *de novo* biosynthesis of histidine to sustain growth, but a histidine auxotroph fails to grow.” (Page- 4; Line- 57-62).

11. P5L67 Replace coherence with concordance

>> This has been duly updated in the revised manuscript (Page- 5; Line- 77).

12. P5, L70-72 Rewrite this sentence

>> This statement has been rephrased in the revised manuscript as “However molecular histidine is abundantly present in mammalian cells (mouse plasma contains $\sim 33 \mu\text{M}$ of histidine, totalling upto $\sim 10 \mu\text{g}$ in 2 ml of mouse blood), suggesting that *Mtb* may never

have to experience histidine auxotrophy in its natural environment within the host.” (Page- 5; Line- 81-84).

13. Infection pervasiveness??? Rewrite

>> The sentence containing this phrase has been rephrased in the revised manuscript as “Nonetheless, we determined the survivability of H37Rv, *ΔhisD* and *chisD* strains within mouse and human monocytic cell lines, Raw264.7 and THP1, respectively.” (Page- 5; Line- 84-85).

14. P7, L125: You write “Histidine is an abundantly available resource in the host cellular milieu, and is readily available for the bacillus for uptake, unless the host has its own mechanism to restrict histidine for example through efficient sequestration”. Whats the evidence for this? This sentence needs a reference? Intracellularly? Extracellularly??? Where in the host are you talking about?

>> Molecular histidine is abundantly present in mammalian cells, for example mouse plasma contains ~33 μM of histidine, totalling upto ~10 μg in 2 ml of mouse blood.

>> The study from Takach E. *et al.* presents the quantitation of free amino acids in the difference organs from mice and rat. The study presents the abundance of amino acids in both intracellular as well as extracellular compartments for various tissues.

>> In addition, the study by Hu H. *et al.* reports the abundance of free histidine with concentration ranging from ~20 μM (starved) to ~ 150 μM (normal) in HeLa cells.

>> This information has been duly updated in the text (Page- 5; Line- 81-84) (Page- 8; Line- 141-143) and the aforementioned references (Reference no. 20 & 21) are incorporated.

15. P8, 128-133. Please rewrite. This doesn't quite make sense

>> This section has been rephrased in the revised manuscript as “In this respect we examined the KEGG Pathways for the probable histidine sequestering/catabolizing enzymes in mammals. The results suggested two enzymes involved in the catabolism of histidine- HAL and HDC. HAL catalyses the conversion of histidine to ammonia and urocanate. HDC catalyses the decarboxylation of histidine to histamine. While both the aforementioned enzymatic reactions are reversible in nature, the *Mtb* genome does not possess genes homologous to HAL and HDC. This implicates that the bacillus isn't capable of regenerating histidine from either urocanate or histamine.” (Page- 8; Line- 144-150).

Reviewers' comments:

Reviewer #1 (Remarks to the Author):

The authors have addressed most of the points raised previously, and the manuscript is much improved as a result. However, a few points remain that need to be addressed:

The measurement of the peak areas in the LC-MS assays for histidine and tryptophan shown in Supplementary Figures 19 and 20 requires further revision for consistency and accuracy. For example, in Figure S19, in some samples, the peak areas are measured between retention times of ~0.45 minutes to ~1.45 minutes, whereas in others it is only measured until 0.7 minutes, and in one case only from 0.6 minutes. A similar problem is occurring in Figure S21, where there is a serious issue with the peak area calculation failing to follow the drifting baseline correctly. There is also an issue in Figure 21 A and B where standard tryptophan amounts of 20ng and 80ng both appear to be estimated at ~54ng. There are several obvious errors in this Figure that need to be corrected before these results can be considered believable: "20 ng" is an amount, not a concentration - this error is also found in Figure S18; the x-axis in Figure 21 A is unlabelled, etc.

2) If "log fold" just means "10 fold", then why not say the latter? It would be much clearer.

3) The clarity of language is improved, but could still benefit from some copy-editing in places e.g. Line 426: Is analogize the correct verb? Would "correlate" or "corroborate" be better choices? Lines 40-43: Tautological use of essential/essentiality.

Reviewer #2 (Remarks to the Author):

The revised manuscript "De novo histidine biosynthesis protects Mycobacterium tuberculosis from host IFN- γ mediated histidine starvation" by Abhisek Dwivedy et al is an interesting and novel study exploring the interaction between host and pathogen metabolism, specifically histidine metabolism and immunology and therefore adds to a body of research in the emerging field of immunometabolism. As such this is an important study which advances the TB field. This revised version is much improved and has answered most of all my previous concerns. Very minor points below.

Minor comments

1. The first two sentence in the abstract need changing. Too clunky and long. The whole abstract could do with sharpening up.
2. I still don't understand why you are referencing histidine levels in serum as evidence for Mtb not being subjected to reduced histidine. Mtb doesn't live in serum. It spends most of its time in phagosomes. Is there actual evidence that phagosomes have plentiful histidine? Im not sure there is. You don't seem to have a reference for this.
3. L149 Change this implicates to this implies?
4. The increase in expression of HDC is often correlated to an increased production of 159 histamine, a well-known mediator of inflammation (this needs a reference)

Authors' responses to the reviewer's comments are written in bold.

Reviewer #1 (Remarks to the Author):

The authors have addressed most of the points raised previously, and the manuscript is much improved as a result.

>> We thank you for the in-depth analysis of our study.

However, a few points remain that need to be addressed: The measurement of the peak areas in the LC-MS assays for histidine and tryptophan shown in Supplementary Figures 19 and 20 requires further revision for consistency and accuracy. For example, in Figure S19, in some samples, the peak areas are measured between retention times of ~0.45 minutes to ~1.45 minutes, whereas in others it is only measured until 0.7 minutes, and in one case only from 0.6 minutes. A similar problem is occurring in Figure S21, where there is a serious issue with the peak area calculation failing to follow the drifting baseline correctly. There is also an issue in Figure 21 A and B where standard tryptophan amounts of 20ng and 80ng both appear to be estimated at ~54ng. There are several obvious errors in this Figure that need to be corrected before these results can be considered believable: "20 ng" is an amount, not a concentration - this error is also found in Figure S18; the x-axis in Figure 21 A is unlabelled, etc.

>> To the best of our knowledge, this is the first ever study to quantitate the levels of free molecular histidine and tryptophan in mice lungs using mass spectrometry. In order to achieve this, we had to first develop a Selective Reaction Monitoring (SRM) method. For method development, linearity curve generation and quality control analysis of the developed method, analytical grade histidine and tryptophan were used.

>> For histidine, the precursor ion with an m/z value of 156.312 produced for daughter ions with m/z values of 55.899, 83.113, 93.101 and 110.012. For method development, the program Thermo trace finder was tuned for automatic peak integration based on the nearest peak criteria specifically for the daughter ion with an m/z value of 110.012, which had a retention time of ~0.5 min. While performing quantitation of histidine in the lung samples, the software integrated the peaks corresponding only to the aforementioned daughter ion. Thus for integration of peaks in samples B6-H37Rv-Day1 and B6-H37Rv-Day15 (Please refer to Supplementary Figure 19), the software selected the region between the retention times of 0.5 min to 1.4 min corresponding to the m/z of 110.012 and calculated the area under the selected region. However, for the sample B6-H37Rv-Day28 (Please refer to Supplementary Figure 19), the software only selected the region between the retention times of 0.5 min to 0.8 min as only those ions had an m/z value of 110.012.

>> Similarly for tryptophan quantitation, the precursor ion had an m/z value of 205.03 and the daughter ions had m/z values of 118.042, 146.071 and 188.071. Of these the daughter ion with an m/z value of 188.071 with a retention time of ~0.8 min was considered for method development.

>> For clarity, we have included the above information in the methodology section (Page No: 32, Line No: 707-712).

>> We apologize for the unintentional typographical errors in Supplementary Figure 21 B. The concentrations presented as 53.448 and 54.151 actually represent 23.448 and 84.151, and the same have been updated in the revised manuscript.

>> The values presented in Supplementary Figure 18 A and C, and Supplementary Figure 21 B represent concentration and the unit has been updated to ng/ml.

>> The unit in x-axis has been updated to ng/ml in Supplementary Figure 21 A.

2) If “log fold” just means “10 fold”, then why not say the latter? It would be much clearer.

>> We have updated this nomenclature in the revised manuscript. (Page No: 6, Line No: 98; Page No: 13, Line No: 271-272; Page No: 15, Line No: 309-310)

3) The clarity of language is improved, but could still benefit from some copy-editing in places e.g. Line 426: Is analogize the correct verb? Would “correlate” or “corroborate” be better choices? Lines 40-43: Tautological use of essential/essentiality.

>> The manuscript was subjected to a second round of proof-reading using Grammarly. We have appropriately re-phrased sentences at various places in the revised version which has further improved the readability of the manuscript. The changes in text are highlighted in yellow background.

>> The sentence containing the word “analogize” has been re-phrased in the updated manuscript. (Page No: 20, Line No: 426-429)

>> The phrase “essentiality of the” has been removed from the sentence presented in Page No: 3, Line No: 40-42.

Authors' responses to the reviewer's comments are written in bold.

Reviewer #2 (Remarks to the Author):

The revised manuscript “De novo histidine biosynthesis protects *Mycobacterium tuberculosis* from host IFN- γ mediated histidine starvation” by Abhisek Dwivedy et al is an interesting and novel study exploring the interaction between host and pathogen metabolism, specifically histidine metabolism and immunology and therefore adds to a body of research in the emerging field of immunometabolism. As such this is an important study which advances the TB field. This revised version is much improved and has answered most of all my previous concerns. Very minor points below.

>> We thank you for the in-depth analysis of our study.

Minor comments

1. The first two sentence in the abstract need changing. Too clunky and long. The whole abstract could do with sharpening up.

>> The manuscript was subjected to a second round of proof-reading using Grammarly. We have appropriately re-phrased sentences at various places in the revised version which has further improved the readability of the manuscript. The changes in text are highlighted in yellow background.

>> We have attempted to re-structure the abstract in order to improve its readability. In our view, this has immensely improved the quality of the manuscript.

2. I still don't understand why you are referencing histidine levels in serum as evidence for Mtb not being subjected to reduced histidine. Mtb doesn't live in serum. It spends most of its

time in phagosomes. Is there actual evidence that phagosomes have plentiful histidine? I'm not sure there is. You don't seem to have a reference for this.

>> We have provided references for the concentration of free molecular histidine in mouse plasma (Reference No: 20 & 21) (Page No: 5, Line No: 81) to provide an impression about the abundance of histidine in mammalian systems.

>> We agree that *Mtb* doesn't live in serum. However, we would like to emphasize that the quantitation of histidine presented in this study is of mouse lungs (a sum total of all intracellular and extracellular free histidine). While the extracellular histidine is unavailable to *Mtb*, the intracellular histidine pools of the *Mtb* infected macrophages and dendritic cells of lungs are an abundant source of the amino acid. This study presents the first ever quantitation of total histidine concentration in mice lungs.

>> In addition, recent studies have demonstrated the various mechanism by which *Mtb* captures and utilizes host amino acids while residing within the phagosomes (Reference No: 22 & 23) (Page No: 5, Line No: 83-84). This further suggests that intracellular amino acids are readily available for uptake by *Mtb* while still residing within the phagosomes, thus a decrease in the intracellular amino acids levels ultimately starves *Mtb* of these vital resources.

3. L149 Change this implicates to this implies?

>> The suggestion has been duly implemented in the revised manuscript. (Page No: 8, Line No: 150)

4. The increase in expression of HDC is often correlated to an increased production of histamine, a well-known mediator of inflammation (this needs a reference)

>> Appropriate references in support of the aforementioned statement have been duly incorporated in the revised manuscript (Reference No: 29 & 30) (Page No: 8, Line No: 160).

Reviewers' comments:

Reviewer #2 (Remarks to the Author):

I am now happy that the authors have addressed all of my comments. This is a very interesting body of work which makes an important contribution to our understanding of our understanding the immunometabolic cross talk between Mtb and its host.

Reviewer's Comments:

"The measurement of the peak areas in the LC-MS assays for histidine and tryptophan shown in Supplementary Figures 19 and 20 requires further revision for consistency and accuracy. For example, in Figure S19, in some samples, the peak areas are measured between retention times of ~0.45 minutes to ~1.45 minutes, whereas in others it is only measured until 0.7 minutes, and in one case only from 0.6 minutes. A similar problem is occurring in Figure S21, where there is a serious issue with the peak area calculation failing to follow the drifting baseline correctly."

Authors' responses

>> We thank the reviewer for the constructive comment.

>> As suggested, we reacquired and reanalysed the data sets for histidine and tryptophan quantitation using a different software- Thermo Xcalibur.

>> For urocanate, histamine and methyl-histidine, like that in the previous version, we are presenting the relative abundance of these metabolites at different time points of infection with respect to day 1 and not the actual quantitative values. As described in the methods sections, the relative abundance of a given metabolite was first normalized with the quantitated value for histidine from the same sample, followed by weight normalization (Page 33, Line 721-726). Thus, for quantitation of metabolites, the new results obtained for histidine quantitation were taken as a reference.

>> The new results correlate better with our hypothesis. As observed in case of B6 mice infected with Δ hisD, the cfu counts start declining from day 21 and there comparatively lesser expression for HAL and HDC as compare to B6 mice infected with H37Rv. The new quantitative data suggest that in H37Rv infected mice the histidine sequestration is more pronounced as compared to Δ hisD infected mice.

>> We have incorporated the new results in Figure 6 and Supplementary Figures 18, 19 and 21 of the updated manuscript.

>> Below we present a comparison of the new and old analysis, pertaining to methods development, quantitation and dynamics of the amino acids and metabolites.

>> Taken together, no significant variation between the new and the previous results was observed.

>> All changes incorporated in the manuscript text are highlighted in yellow in the file submitted as "Manuscript marked up".

P.T.O.

METHOD DETAILS

	New analysis (presented in updated manuscript)	Original analysis (presented in previous version)
Software	Thermo Xcalibur	Thermo Trace Finder
Baseline correction	Manual, Peak to peak noise calculation, Tolerance set to 0.01 and 0.05 for histidine and tryptophan, Signal to noise set to 80 and 50 for histidine and tryptophan, Edge flattening enabled, Baseline scan set to 30	Automated, Peak to peak noise calculation
Peak Selection	Manual, nearest peak criteria based on retention time, Peak Signal to noise cutoff set to 30 and 10 for histidine and tryptophan	Automated, nearest peak criteria based on m/z scanning coupled with retention time
Peak Integration	Manual, fixed length	Automated, m/z scanning

>>The details of the methodology has been incorporated in the updated manuscript (Page 33, Line 714-722).

P.T.O.

HISTIDINE QUANTITATION

Method Development		
	New analysis (presented in updated manuscript)	Original analysis (presented in previous version)
Reproducibility- 1			Concentration: 500 ng/ml Retention time: ~ 0.55 s Area: 846394	Concentration: 500 ng/ml Retention time: ~ 0.55 s Area: 1056690.912
Reproducibility- 1			Concentration: 500 ng/ml Retention time: ~ 0.55 s Area: 821405	Concentration: 500 ng/ml Retention time: ~ 0.55 s Area: 1021159.459
Linearity Curve		
Quality Control 1			Calculated concentration- 50 ng/ml Experiment determined concentration- 51.56 ng/ml	Calculated concentration- 50 ng/ml Experiment determined concentration- 51.767 ng/ml
			Calculated concentration- 200 ng/ml Experiment determined concentration- 212.67 ng/ml	Calculated concentration- 250 ng/ml Experiment determined concentration- 238.659 ng/ml

B6 mice infected with H37Rv		
	New analysis (presented in updated manuscript)	Original analysis (presented in previous version)
Day 1		

B6 IFN- γ ^{-/-} mice infected with H37Rv

TRYPTOPHAN QUANTITATION

Method Development		
	New analysis (presented in updated manuscript)	Original analysis (presented in previous version)
Linearity Curve	$Y = 1912 \cdot X + 10980$ $R^2 = 0.9807$ 	TRYPTOPHAN $Y = 5328.08 + 2036.78 \cdot X$ $R^2 = 0.9970$ W: Equal Quality Control 1			Calculated concentration- 20 ng/ml Experiment determined concentration- 20.27 ng/ml	Calculated concentration- 20 ng/ml Experiment determined concentration- 23.448 ng/ml
			Calculated concentration- 80 ng/ml Experiment determined concentration- 87.51 ng/ml	Calculated concentration- 80 ng/ml Experiment determined concentration- 84.151 ng/ml

B6 mice infected with H37Rv

	New analysis (presented in updated manuscript)	Original analysis (presented in previous version)

B6 IFN- γ ^{-/-} mice infected with H37Rv

New analysis (presented in updated manuscript)

Original analysis (presented in previous version)

Tryptophan dynamics over 4 weeks of infection

	New analysis (presented in updated manuscript)	Original analysis (presented in previous version)
--	--	---

P.T.O.

METABOLITES

Metabolite dynamics over 4 weeks of infection		
	New analysis (presented in updated manuscript)	Original analysis (presented in previous version)
Urocanate	Relative change in abundance (w.r.t day1, Urocanate)	Relative change in abundance (Urocanate)
Histamine	Relative change in abundance (w.r.t day1, Histamine)	Relative change in abundance (Histamine)
Methyl-histidine	Relative change in abundance (w.r.t day1, Methyl-histidine)	Relative change in abundance (Methyl-histidine)

| IFN- γ ^{-/-} B6:H37Rv
 | B6:H37Rv